# Learning to Plan Like the Human Brain via Visuospatial Perception and Semantic-Episodic Synergistic Decision-Making

**Tianyuan Jia**
Beijing Normal University
Beijing Institute of Technology
tianyj@mail.bnu.edu.cn

**Ziyu Li**
Beijing Institute of Technology
ziyuli@bit.edu.cn

**Qing Li**[*]
Beijing Institute of Technology
liqing@bit.edu.cn

**Xiuxing Li**
Beijing Institute of Technology
xxl@bit.edu.cn

**Xiang Li**
Li Auto Inc.
lixiang39@lixiang.com

**Wei Chen**
Li Auto Inc.
chenwei10@lixiang.com

**Li Yao**
Beijing Normal University
yaoli@bnu.edu.cn

**Xia Wu**
Beijing Institute of Technology
wuxia@bit.edu.cn

## Abstract

Motion planning in high-dimensional continuous spaces remains challenging due to complex environments and computational constraints. Although learning-based planners, especially graph neural network (GNN)-based, have significantly improved planning performance, they still struggle with inaccurate graph construction and limited structural reasoning, constraining search efficiency and path quality. The human brain exhibits efficient planning through a two-stage Perception-Decision model. First, egocentric spatial representations from visual and proprioceptive input are constructed, and then semantic–episodic synergy is leveraged to support decision-making in uncertainty scenarios. Inspired by this process, we propose NeuroMP, a brain-inspired planning framework that learns to plan like the human brain. NeuroMP integrates a Perceptive Segment Selector inspired by visuospatial perception to construct safer graphs, and a Global Alignment Heuristic guide search in weakly connected graphs by modeling semantic-episodic synergistic decision-making. Experimental results demonstrate that NeuroMP significantly outperforms existing planning methods in efficiency and quality while maintaining a high success rate.

## 1 Introduction

Motion planning is fundamental to robotic systems, with broad applications in autonomous vehicles [1], logistics [2], and so on. Its primary objective is to compute a high-quality, collision-free path in the configuration space that connects the start and goal [3]. However, real-world configuration spaces often involve numerous continuous variables and complex high-dimensional properties, significantly increasing the difficulty of the planning problem. Classical motion planners are typically categorized as search-based or sampling-based. Search-based methods, such as A* [3] and Dijkstra [4], formulate planning as a graph search. While sampling-based methods, such as RRT [5] and its variants [6, 7],

---

[*]Corresponding Author

39th Conference on Neural Information Processing Systems (NeurIPS 2025).

construct paths by randomly sampling the configuration space. However, these methods suffer from a rapid increase in complexity in high-dimensional continuous spaces [8].

To enhance planning efficiency and quality, researchers have proposed learning-based planners that integrate data-driven models with classical planning methods. By leveraging the learning and representation capabilities of deep neural networks, these methods can extract key patterns from the configuration space or learn the behavior of oracle planners, thereby optimizing critical steps in the planning process to enhance overall performance [9]. Various neural-network-driven motion planners have been proposed, including convolutional neural networks (CNNs) [10–12], recurrent neural networks (RNNs) [13–15], and graph neural networks (GNNs)[9, 16, 17]. Among these, GNN-based methods demonstrate significant advantages in processing high-dimensional graph tasks. Specifically, GNN-based methods enhance search efficiency by operating on random geometric graphs (RGGs) constructed from sampled configurations, avoiding full workspace encoding [16]. Here, RGGs denote the undirected k-nearest neighbor (k-NN) graph built on free configuration space. They also predict node or edge priorities via graph pattern learning, reducing redundancy and improving path exploration quality.

GNN-based planners, such as GNN-Explorer [9] and GraphMP [18], demonstrate strong performance but still exhibit notable limitations. GNN-Explorer prioritizes edges without considering total path costs, limiting optimal path selection. GraphMP addresses this issue by incorporating a Neural Collision Checker to remove predicted collision edges and embedding GNN-based heuristics within an A* search framework. However, GNN-based planners still face two key limitations: (i) Inaccurate graph construction. Neural Collision Checker may incorrectly retain collision edges or remove valid ones, disrupting heuristic estimation. (ii) Limited structural reasoning. Excessive edge removal weakens the structural cues essential for reasoning, degrading information propagation and representation. This issue stems from the reliance on local feature aggregation and neighborhood-based message passing, which are insufficient to capture global structural information, thereby limiting the representational capacity of models. Although BrainyMP [19] introduces subgraph structures to enhance representation, it remains limited by subgraph coverage and redundant computation.

In contrast, humans exhibit remarkable capabilities in planning tasks, especially in uncertain or incomplete scenarios. The cognitive process can be divided into two functional stages: *perception* and *decision-making* [20, 21]. The first stage is *visuospatial perception*, where the brain initially acquires external environmental information through the visual system. The visual cortex encodes and processes raw visual signals from the retina, identifying key features such as obstacles, boundaries, and feasible regions [21]. Subsequently, the posterior parietal cortex (PPC) transforms visual and proprioceptive information into egocentric spatial representations useful for planning [22, 23]. The second stage is *semantic–episodic synergistic decision-making*, where the brain engages the semantic–episodic synergy mechanism to support environmental reasoning under weak or ambiguous conditions [24–26]. Specifically, episodic and semantic information is integrated through coordinated activity of the prefrontal cortex (PFC), anterior insula (AI), and default mode network (DMN), triggering a shared cognitive control mechanism [24, 27]. The global semantic framework provides contextual support for episodic memory, aiding decision-making under uncertain cues. While episodic memory refines or corrects the semantic framework with specific event details. Accordingly, we model the brain's perception and decision-making processes, as illustrated in Fig. 1, offering valuable insights for improving motion planning systems.

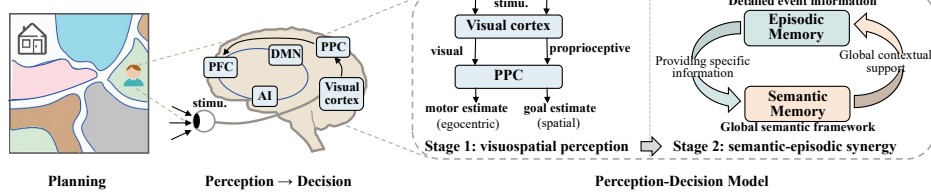

Figure 1: The two-stage Perception-Decision model. The cognitive process of planning consists of two functional stages. Stage 1: Visuospatial perception inspires the extraction of environmental features and their interrelations for subsequent planning. Stage 2: Semantic–episodic synergistic decision-making provides two key insights: (i) incorporating global topological constraints to enrich representations, and (ii) bidirectional complementarity between complete graphs and weak graphs.

Inspired by the two-stage Perception–Decision model, we propose **NeuroMP**, a brain-inspired motion planning framework to address the limitations of existing GNN-based planners. In the first stage,

mimicking the *visuospatial perception* process, the **Perceptive Segment Selector** efficiently identifies and reasons environmental patterns and their spatial relationships to construct safer graphs. In the second stage, motivated by *semantic–episodic synergistic decision-making*, the **Global Alignment Heuristic** is proposed to improve structural reasoning. Specifically, drawing on the contextual understanding of the global knowledge framework in semantic memory, the module incorporates spectral features to capture global topological constraints. Furthermore, inspired by the bidirectional interaction process, the **Alignment Dual-Flow Learning** strategy leverages the complete graph to guide heuristic estimation in weakly connected graphs via dual-channel collaborative learning. The proposed method is evaluated on maze and robotic-arm manipulation tasks, with experimental results demonstrating improved planning efficiency and path quality while maintaining high reliability.

## 2 Related Work

**Classical planning methods.** Current classical motion planners are primarily divided into two popular categories: search-based and sampling-based planners. Search-based planners typically perform iterative searches on RGGs to find optimal paths, such as BFS [28] and Dijkstra [4]. These methods are widely used in low-dimensional discrete scenarios but become inefficient in high-dimensional environments due to high computational complexity [6]. To overcome this limitation, informed search strategies have been proposed, such as A* [3] and its variants [29–31], which significantly reduces the search space and enhances efficiency through heuristic functions. Nevertheless, these methods are typically limited to discrete spaces, and their search efficiency still suffers from exponential complexity in high-dimensional environments.

In contrast to search-based methods, sampling-based methods find solutions by randomly sampling nodes in configuration spaces, such as RRT* [6] and its variants [7]. These methods avoid explicit discretization and construction of entire spaces, alleviating computational complexity. However, they often struggle to identify critical regions and efficiently sample sparse states accurately. To address these issues, heuristic-enhanced sampling-based methods have been proposed, such as BIT* [32] and LazySP [33]. Despite these improvements, sampling-based methods yield only near-optimal solutions and exhibit limited performance in high-dimensional complex environments [10].

**Learning-based planning methods.** By leveraging the powerful representation capabilities of neural networks, learning-based methods have revitalized traditional search- and sampling-based planners, significantly improving planning efficiency. Learning-based search methods employ neural networks to generate enhanced heuristic functions or directly learn planning policies. For example, VIN [34] and SPT [13] directly learn search paths, reducing search space and time. Neural A* [18] reformulates A* as differentiable for end-to-end training in 2D spaces. Additionally, imitation learning has also been integrated into heuristic strategies. For instance, SAIL [35] and TransPath [36] learn heuristics from environment representations using deep neural networks. However, these methods often rely on handcrafted heuristics and are limited to 2D workspaces.

Learning-based sampling planners have substantially improved efficiency and transferability by optimizing sampling strategies or directly generating paths. Representative approaches include MPNet [37], M$\pi$Net [38], and NeuralMP [39], which learn informative samples from environment and configuration data to accelerate planning. OracleNet [40] leverages imitation learning to replicate oracle behaviors; CVAE [8] learns sampling distributions to generate samples along solution paths; and NEXT [10] embeds high-dimensional state spaces into lower-dimensional representations and employs CNNs to learn local sampling strategies. While MPT [41] employs a Transformer to model configuration–environment relations and directly produce feasible paths from historical data and scene context, NTFields [42] learns neural potential fields to generate trajectories, and P-NTFields [43] adopts a probabilistic formulation to handle uncertainty. In addition, dictionary-learning methods [44] optimize sampling dictionaries to enhance efficiency and generalization. RDT-RRT [12] and NIRRT* [11] refine sampling using curvature-aware CNNs and point-Net. Despite these advances, two challenges persist: (i) limited sample quality and collision-avoidance reliability, which can cause redundant exploration or low-quality solutions; and (ii) reliance on dense workspace encoding and extensive perceptual features, which incur substantial computational and memory costs.

GNNs have demonstrated superior performance owing to their strong capability in learning graph patterns and adapting to high-dimensional spaces. For example, GNN-Explorer [16] enhances planning efficiency and robustness by prioritizing the exploration of promising edges. GraphMP [9]

combines GNNs with a differentiable A* module to identify optimal paths over RGGs. Inspired by spatial and relational memory mechanisms, BrainyMP [19] enriches representations with subgraph structures to improve planning performance. However, its ability to capture global information is inherently constrained by the number and coverage of subgraphs, which still limits structural reasoning. Integrating brain-inspired mechanisms more directly into GNN architectures is a promising direction for advancing motion planning.

## 3 Preliminaries

**Task setting.** A motion planning problem in a $d$-dimensional continuous configuration space $C \subseteq \mathbb{R}^d$, where the configuration space is divided into the obstacle space $C_{obs} \subseteq C$ and the free space $C_{free} = C \setminus C_{obs}$. The planning problem is typically formulated as the search process on the $\mathcal{G} = (\mathcal{V}, \mathcal{E})$, where the set of nodes $\mathcal{V}$ is sampled from $C_{free}$, and weighted edges $\mathcal{E}$ are constructed using K-nearest neighbors (K-NN). The start and goal nodes are denoted as $v_s, v_g \in \mathcal{V}$, respectively. The aim is to search a finite set of edges that connects the start and goal nodes within the graph, denoted as $\xi = e_i : (v_{i-1}, v_i)_{i \in [1,T]}$, where $e_i \in \mathcal{E}$, $v_0 = v_s$, $v_T = v_g$, and $v_i \in \mathcal{V}$ for all $i \in [0, T]$.

**Differentiable A\*.** Differentiable A [18] redefines the traditional A algorithm [3], enabling end-to-end differentiable training. Further details are provided in the Appendix A.1.

## 4 Methodology

### 4.1 Overview

The overall framework of NeuroMP is illustrated in Fig. 2. During training, the binary cross-entropy (BCE) loss minimizes the difference between predicted values and ground truth, improving the accuracy of the Perceptive Segment Selector $\mathcal{N}_S$. The Global Alignment Heuristic $\mathcal{N}_H$ is trained jointly with a differentiable A* module using the Alignment Dual-Flow Learning (A2FL) strategy to optimize heuristic estimation. During online planning, selective sampling is employed to construct the RGG based on the bias probability $\beta$, reducing redundant exploration. Then, $\mathcal{N}_S$ predicts edge collision probabilities and filters unsafe edges to generate a safe RGG'. To address weak connectivity, $\mathcal{N}_H$ incorporates spectral features to impose global topological constraints, enabling more accurate heuristic predictions. Finally, these heuristics are passed to the A* module for path search, followed by shortcut retrieval to remove detours and improve path quality. Details of the selective sampling and shortcut retrieval optimization steps are provided in the Appendix A.2.

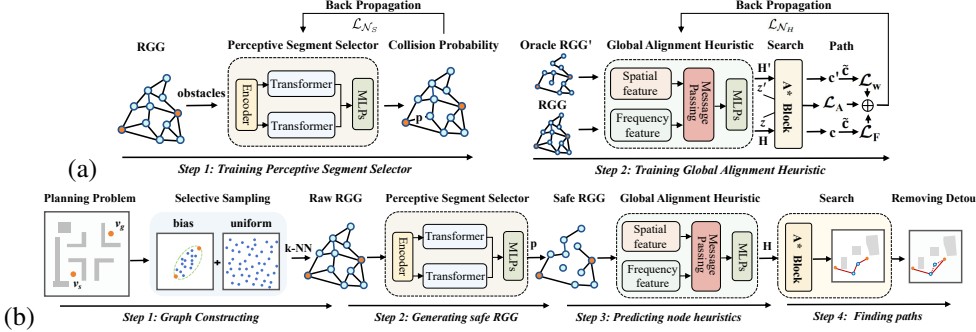

Figure 2: Overview of NeuroMP. (a) The training process. The $\mathcal{N}_S$ and $\mathcal{N}_H$ are trained independently. (b) The online planning process. The learned $\mathcal{N}_S$, $\mathcal{N}_H$, and A* modules are integrated to perform graph search-based planning.

### 4.2 Perceptive Segment Selector

In the first stage, the proposed Perceptive Segment Selector module identifies and reasons about spatial patterns and structural relationships, enhancing collision prediction accuracy and facilitating the construction of safer graphs. The overall architecture of $\mathcal{N}_S$ is illustrated in Fig. 3.

For the input RGG $\mathcal{G} = (\mathcal{V}, \mathcal{E})$, its nodes and edges are first encoded into a latent space with $x \in \mathbb{R}^{|\mathcal{V}| \times d_h}$, $y \in \mathbb{R}^{|\mathcal{E}| \times d_h}$, where $d_h$ is the encoding size. Specifically, the embedding for the $i$-th node $v_i$ and the $l$-th edge $e_{ij}$ are represented as $x_i = f_x(v_i)$ and $y_{ij} = f_y(v_i, v_j, v_i - v_j)$, where $f_x$ and $f_y$ are two separate two-layer MLPs.

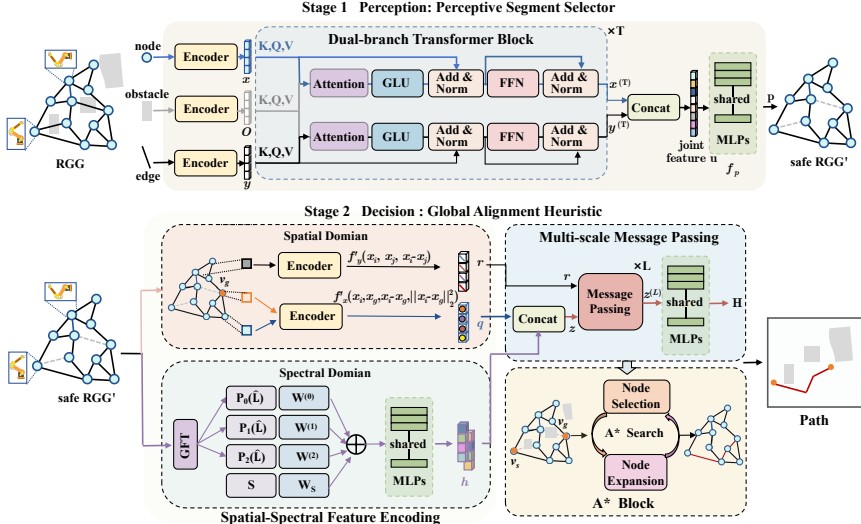

Figure 3: The model architecture. The upper part illustrates the Perceptive Segment Selector, while the lower part presents the Global Alignment Heuristic.

**Obstacle Encoding.** The obstacle information is encoded into the node and edge features. To simplify the notation, the feature embeddings of all nodes and edges are denoted as $x$ and $y$, respectively. The obstacle configuration is denoted as $O \in \mathbb{R}^{|o| \times 2n}$, where $|o|$ represents the variable number of obstacles, $n$ denotes the workspace dimension. In the $t$-th iteration of obstacle encoding, the interactions between the node, edge, and obstacle features are encoded as follows,

$$a_x^{(t)} = x^{(t)} + GLU\left(Att(f_{K_x}^{(t)}(O), f_{Q_x}^{(t)}(x^{(t)}), f_{V_x}^{(t)}(O))\right),$$
$$x^{(t+1)} = \mathbf{LN}(a_x^{(t)} + f_{a_x}^{(t)}(a_x^{(t)})), \text{with } a_x^{(t)} = \mathbf{LN}(a_x^{(t)}),$$
(1)

$$a_y^{(t)} = y^{(t)} + GLU\left(Att(f_{K_y}^{(t)}(O), f_{Q_y}^{(t)}(y^{(t)}), f_{V_y}^{(t)}(O))\right),$$
$$y^{(t+1)} = \mathbf{LN}(a_y^{(t)} + f_{a_y}^{(t)}(a_y^{(t)})), \text{with } a_y^{(t)} = \mathbf{LN}(a_y^{(t)})$$
(2)

where $\mathbf{LN}$ refers to layer normalization, $f_{K_x}, f_{Q_x}, f_{V_x}, f_{a_x}, f_{K_y}, f_{Q_y}, f_{V_y}, f_{a_y}$ represent different MLPs layers. The Gated Linear Unit (GLU) is defined as $GLU(A) = \sigma(W_a A + b_a) \odot (W_h A + b_h)$, $W_a, W_h$ are learnable weight matrices, $b_a, b_h$ are bias vectors. $Att(K, Q, V)$ denotes the weighted results of self-scores and cross-scores through the gating mechanism. For the obstacle encoding of nodes, the attention mechanism can be computed as follows,

$$Att(K, Q, V) = \text{softmax}\left(\frac{[Q_n K_o^T, Q_n K_n^T]}{\sqrt{d_k}}\right) \odot [V_n, V_o]$$
(3)

where the $\odot$ denotes the element-wise product, $[Q_n, K_n, V_n]$, $[Q_o, K_o, V_o]$ represent the keys, queries, and values for the nodes and obstacles, respectively. The obstacle encoding of edges follows a similar process.

**Collision Probability Prediction.** Subsequently, the information of the graph and obstacles is encoded into a joint embedding vector $\mathbf{u}_{ij}$ for the feature embedding of each edge $e_{ij}$. After $T$ iterations of obstacle encoding, $\mathbf{u}_{ij}$ is represented as $\mathbf{u}_{ij} = (x_i^{(T)}, x_j^{(T)}, x_i^{(T)} - x_j^{(T)}, y_{ij}^{(T)})$. The collision probability $\mathbf{p}_{ij}$ for each edge $e_{ij}$ is computed through a three-layer MLP $f_p$, i.e., $\mathbf{p}_{ij} = f_p(\mathbf{u}_{ij})$.

**Learning Procedure.** Each training instance $\{\mathcal{G}, C_{obs}\}^{(i)}$ consists of a graph $\mathcal{G} = (\mathcal{V}, \mathcal{E})$ sampled within the configuration space and a set of obstacles $C_{obs}$, where $\mathcal{V}$ represents all the sampled nodes,

$\mathcal{E}$ represents the edges constructed based on K-NN, and $v_s, v_g \in \mathcal{V}$. To optimize the model $\mathcal{N}_S$ for predicting the collision probability of each edge, a BCE loss is employed to minimize the gap between the predicted probability vector and the ground-truth labels. The loss function is defined as:

$$\mathcal{L}_{\mathcal{N}_S} = -(\widetilde{\mathbf{p}}_{ij} \log \mathbf{p}_{ij}) + (1 - \widetilde{\mathbf{p}}_{ij})(\log(1 - \mathbf{p}_{ij})) \tag{4}$$

where $\widetilde{\mathbf{p}}_{ij}$ is determined by the Dijkstra algorithm, which labels each edge as 1 if no collision occurs and 0 if a collision is detected.

## 4.3 Global Alignment Heuristic

In the second stage, the Global Alignment Heuristic incorporates spectral features to capture global topological constraints, enriching node representations. Furthermore, we propose a dual-channel collaborative learning strategy, called Alignment Dual-Flow Learning (A2FL). This strategy allows the complete graph to provide shared semantic representations to the weakly connected graph, while the weakly connected graph offers structural feedback to refine representations in the complete graph, thus improving heuristic estimation in weakly connected graphs. The network architecture of $\mathcal{N}_H$ is shown in Fig. 3.

### 4.3.1 Spatial-Spectral Feature Encoding

Traditional node and edge feature embeddings rely on local features, limiting the ability to capture the global topological structure. To overcome this limitation, we introduce spectral features derived from the graph Laplacian matrix to capture the global topological characteristics. These features are then integrated into the node embedding, enhancing the model's graph representation capability.

**Spatial Feature Encoding.** Given the safe RGG$'$ $\mathcal{G}' = (\mathcal{V}', \mathcal{E}')$ and the goal $v_g$ as input, the nodes and edges of the $\mathcal{G}'$ are similarly encoded into a latent space. Specifically, the feature embedding of the robot state $x_i$ is computed by incorporating the difference and the L2 distance to the goal state $x_g$, i.e., $q_i = f'_x(x_i, x_g, x_i - x_g, \|x_i - x_g\|_2^2)$, and the feature embedding of $l$-th edge $e_{ij}$ is defined as $r_{ij} = f'_y(x_i, x_j, x_j - x_i)$, where $f'_x$ and $f'_y$ are two different two-layer MLPs.

**Spectral Feature Encoding.** To enhance the node feature representation, we extract spectral features using the Graph Fourier Transform (GFT) to obtain filters through the eigendecomposition of the Laplacian matrix. The Laplacian matrix is defined as $\hat{L} = I - \hat{A} = U\Lambda U^T$, where $\hat{A}$ is the normalized adjacency matrix, $I$ is the identity matrix, $U$ is the matrix of eigenvectors, and $\Lambda$ is the diagonal matrix of eigenvalues. Subsequently, node features recursively propagate through the Laplacian matrix $\hat{L}$. The spectral feature of each node $v_i$ after $K$-order propagation is computed as follows,

$$h_i = \sum_{k=0}^{K} P_k(\hat{L}) x_i W^{(k)} + SW_s \tag{5}$$

where $P_k(\hat{L})$ denotes the $k$-th order polynomial expansion of the Laplacian matrix, $S$ represents another feature subspace generated by the principal components of the structure matrix, and learnable weights $W^{(k)}$ and $W_s$ enable flexible re-weighting of each feature subspace.

### 4.3.2 Multi-scale Message Passing

The spectral feature is concatenated with the spatial feature into an embedding vector $z_i = (q_i, h_i)$. The node and edge embeddings are then iteratively updated by aggregating the local information of each node from its neighbors $\mathcal{N}(v_i) = \{v_j | e_{ij} \in \mathcal{E}'\}$,

$$\begin{aligned}
a_i^{(l)} &= \max(\{f_a(z_i^{(l)}, z_j^{(l)}, z_j^{(l)} - z_i^{(l)}, r_{ij}^{(l)}) | v_j \in \mathcal{N}(v_i)\}), \\
z^{(l+1)} &= f_z(z_i^{(l)}, a_i^{(l)}), \forall v_i \in \mathcal{V}' \\
r_{ij}^{(l)} &= \max(r_{ij}^{(l)}, f_r(z_i^{(l)}, z_j^{(l)}, z_i^{(l)} - z_j^{(l)})), \forall e_{ij} \in \mathcal{E}'
\end{aligned} \tag{6}$$

where $f_a$, $f_z$, and $f_r$ are three different two-layer MLPs. After $L$ iterations, the heuristic value of node $v_i$ is computed as $\mathbf{H}_i = f_H(z_i^{(L)})$, where $f_H$ is a three-layer MLP.

### 4.3.3 Alignment Dual-Flow Learning

To enhance the heuristic estimation capability of $\mathcal{N}_H$, we introduce the Alignment Dual-Flow Learning (A2FL) strategy to train $\mathcal{N}_H$, as illustrated in Fig. 4. The original channel provides global knowledge to iteratively refine heuristic learning in the weak channel, while sparse connections in the weak channel also influence updates in the original channel. Subsequently, $\mathcal{N}_H$ is jointly trained with the A* module end-to-end to optimize heuristic estimation.

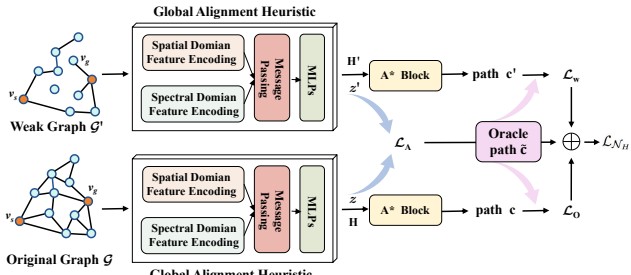

Figure 4: The Alignment Dual-Flow Learning(A2FL) Process.

Given a training problem instance $\{\mathcal{G}, \mathcal{G}', v_s, v_g, \widetilde{\mathbf{c}}\}^{(i)}$, where $\mathcal{G}'$ is the collision-free RGG processed by an oracle planner and the binary vector $\widetilde{\mathbf{c}}$ represents nodes of the oracle path. The training loss for each channel is calculated as the L1 distance between the closed list vector obtained from the A* and the oracle planner, denoted as follows:

$$
\begin{aligned}
\mathcal{L}_W &= \|\widetilde{\mathbf{c}} - \mathbf{c}'\|_1 / |\mathcal{V}'|, \mathcal{V}' \subseteq \mathcal{G}' \\
\mathcal{L}_O &= \|\widetilde{\mathbf{c}} - \mathbf{c}\|_1 / |\mathcal{V}|, \mathcal{V} \subseteq \mathcal{G}.
\end{aligned}
\tag{7}
$$

where the binary vectors $\mathbf{c}$ and $\mathbf{c}'$ mark the nodes in the found path based on the original graph $\mathcal{G}$ and the weakly connected graph $\mathcal{G}'$, respectively. The loss penalizes excessively explored nodes, guiding the search toward the optimal path.

To ensure semantic consistency between the two channels, an alignment loss is introduced to align the latent embeddings $Z = f_n(z)$ and $Z' = f'_n(z')$ derived from node features $z$, $z'$ of the two channels, where $f_n$ and $f'_n$ are two separate MLPs. The alignment loss $\mathcal{L}_A$ is constructed by using positive contrastive difference (the similarity of node features across the two channels) and negative contrastive difference (non-diagonal features within and between channels).

$$
\mathcal{L}_A = -\frac{1}{2|\mathcal{V}|} \sum_{i=1}^{|\mathcal{V}|} \left( \log \frac{f(Z_i, Z'_i)}{\sum_{j \neq i} f(Z_i, Z'_j)} + \log \frac{f(Z_i, Z'_i)}{\sum_{j \neq i} f(Z_j, Z'_i)} \right)
\tag{8}
$$

where $f(a, b) = \exp(\cos(a, b))$, $\cos(a, b)$ is the cosine similarity function. The final objective function is given by $\mathcal{L}_{\mathcal{N}_H} = \mathcal{L}_W + \mathcal{L}_O + \gamma \mathcal{L}_A$, $\gamma$ is used to adjust the magnitude of different losses.

## 5 Experiments

### 5.1 General Setting

**Dataset.** The effectiveness of the proposed method is validated across six types of motion planning tasks with the following environment settings: (i) Stick3: a 3-degree-of-freedom (DoF) stick robot operating in a 2D workspace [10]. (ii) Link8: an 8-DoF link robot in a 2D workspace, sharing the same maze environments as Stick3. (iii) Ur5: a 6-DoF UR5 robot operating in a 3D workspace. (iv) Kuka13: a 13-DoF Kuka robotic arm in a 3D workspace. (v) Kuka2Arms: a dual 7-DoF Kuka robotic arm system in a 3D workspace. (vi) Kuka3Arms: three 7-DoF Kuka robotic arms in a 3D workspace. The Ur5-Kuka2Arms environment is referenced from [16]. Among these environments, Stick3 and Link8 are maze-planning tasks, while the others involve robotic arm manipulation. These two categories represent classic motion planning scenarios, effectively simulating real-world challenges such as obstacle avoidance, path selection, and precise maneuvering.

**Experimental Setting.** For each environment, a training data set of 2000 different planning problems is constructed to train the Perceptive Segment Selector $\mathcal{N}_S$ and the Global Alignment Heuristic $\mathcal{N}_H$ separately. Specifically, each problem instance $\{\mathcal{G}, C_{obs}\}^{(i)}$ consists of the randomly generated RGG and obstacles for $\mathcal{N}_S$. For $\mathcal{N}_H$, each problem instance $\{\mathcal{G}, \mathcal{G}', v_s, v_g, \widetilde{\mathbf{c}}\}^{(i)}$ consists of the start and goal $v_s, v_g$, sampled $\mathcal{G}$, collision-free $\mathcal{G}'$, and the optimal path $\widetilde{\mathbf{c}}$ computed using Dijkstra [4]. Additionally, 500 problem instances are used as a validation data set, where weights of the $\mathcal{N}_H$ with the lowest path cost are retained. After training, an end-to-end evaluation is performed on 1,000 test problems. The proposed method is implemented using PyTorch and executed in an environment equipped with NVIDIA RTX 4070 GPUs. Following GraphMP's parameter settings, the $\mathcal{N}_S$ iterates obstacle encoding three times, with an output dimension of 64. The $\mathcal{N}_H$ employs a message-passing neural network (MPNN) with five iterations and an output dimension of 32. The Adam optimizer [45] is used for training, with 400 epochs, a learning rate of 0.001, and a batch size of 8.

**Baselines.** Seven high-performing baseline planners are selected for comparison to evaluate the performance of NeuroMP comprehensively. These include three classical planners: RRT* [8], BIT* [32], and LazySP [33]; a CNN-based planner: NEXT [10]; two GNN-based planners: GNN-Explorer [16] and GraphMP [9]; and a brain-inspired GNN-based planner: BrainyMP [19], which represents the current state of the art (SOTA). Additionally, GNN-Smoother [16] is employed as a post-processing module for path optimization.

**Evaluation Matrices.** To comprehensively assess the performance of different methods, we select the following evaluation metrics. Specifically, success rate (SR) represents the proportion of successfully planned collision-free paths, indicating the model's reliability. Collision check (CK) denotes the average number of collision checks across test problems, and planning time (PT) measures the total execution time for solving 1,000 test problems. These two metrics assess planning efficiency. Path cost (PC) represents the mean Euclidean length of successfully planned paths, reflecting path quality. Path cost with penalty (PCP) introduces a large penalty for failed cases to ensure a fair comparison across different methods, which is necessary as planners with low success rates typically succeed only in simple tasks with lower costs while failing in more complex tasks that require higher costs.

## 5.2 Overall Performance

Table 1 compares the performance of our method with the baselines across six datasets. Our method achieves the optimal or near-optimal success rate, demonstrating competitiveness with SOTA planners. NeuroMP significantly reduces collision checks compared to GraphMP and BrainyMP across all environments, requiring only 71%, 61%, 57%, 61%, 66%, and 68% of the collision checks performed by GraphMP. This reduction highlights the effectiveness of its edge selection and heuristic estimation. However, a commonly raised issue with learning-based planners is the time overhead associated with the frequent use of large neural network models when solving problems, such as NEXT. However, GNN-based planning methods substantially reduce planning time by searching on the sampled RGG. Compared to the SOTA planners, NeuroMP achieves acceleration across different environments. Classical planners such as RRT*, due to their simplistic design, require minimal planning time but suffer from frequent failures. Another advantage of NeuroMP is its low path cost, which finds high-quality paths across different environments and achieves the lowest path cost in Kuka13 and Kuka2Arms. Although RRT* and NEXT occasionally achieve lower path costs in some environments, their low success rates result in significant penalty costs due to frequent failures. In contrast, the superior success rate of NeuroMP results in very low failure penalties. A comprehensive analysis of the five key evaluation metrics can be found in the Appendix B.

**Collision Prediction Performance.** Table 2 compares the collision probability prediction of the $\mathcal{N}_S$ in NeuroMP and the Neural Collision Checker in GraphMP in various environments, treated as a binary classification task. Evaluation metrics include accuracy, recall, F1 score, and confidence. Higher accuracy, recall, and F1 scores indicate improved collision detection capabilities, while confidence reflects the model's certainty in its predictions. Experimental results demonstrate that $\mathcal{N}_S$ outperforms Neural Collision Checker across all metrics, with particular improvements in high-dimensional environments. This suggests that NeuroMP achieves superior collision detection by effectively reducing false positives and negatives, thereby enhancing the safety and efficiency of motion planning. Moreover, its higher confidence scores indicate more reliable predictions, ensuring stable collision detection support for motion planning.

Table 1: Comparison of the planning performance of all methods across all environments.

| Methods | Stick3 | | | | | Link8 | | | | | Ur5 | | | | |
|---|---|---|---|---|---|---|---|---|---|---|---|---|---|---|---|
| | SR↑ | CK↓ | PC↓ | PCP↓ | PT↓ | SR↑ | CK↓ | PC↓ | PCP↓ | PT↓ | SR↑ | CK↓ | PC↓ | PCP↓ | PT↓ |
| RRT* | 0.62 | 10106.18 | 1.36 | 6.38 | **235.55** | 0.41 | 9332.73 | 7.95 | 26.86 | 1254.93 | 0.39 | 3141.19 | 4.06 | 29.39 | 206.58 |
| BIT* | 0.97 | 11645.68 | 1.70 | 2.13 | 364.18 | **0.93** | 22992.21 | 14.09 | 15.98 | 1350.51 | **1.00** | 5080.90 | 11.20 | 11.35 | 291.74 |
| LazySP | **0.98** | 7719.22 | 1.82 | 2.17 | 891.92 | 0.88 | 10387.94 | 15.81 | 24.15 | 3147.23 | 0.99 | 2699.82 | 12.06 | 12.58 | 482.94 |
| NEXT | 0.96 | 6380.69 | **1.22** | 1.77 | 586.63 | 0.41 | 13022.17 | **7.92** | 26.95 | 19494.83 | 0.37 | 6512.62 | **3.65** | 28.68 | 5232.09 |
| GNN-Explorer | **0.98** | 8805.60 | 2.02 | 2.23 | 687.95 | 0.91 | 10824.59 | 19.53 | 21.29 | 1872.64 | 0.98 | 3184.29 | 12.51 | 12.86 | 585.67 |
| GraphMP | 0.97 | 7485.33 | 1.35 | 1.76 | 491.02 | 0.89 | 13222.98 | 10.59 | 13.80 | 1569.65 | 0.96 | 2706.42 | 8.78 | 10.05 | 266.55 |
| BrainyMP | **0.98** | 6900.51 | 1.39 | 1.68 | 339.63 | 0.90 | 9887.97 | 9.58 | 12.10 | 964.85 | 0.99 | 1639.40 | 7.39 | 7.76 | 160.71 |
| NeuroMP | **0.98** | **5327.99** | 1.32 | 1.62 | 248.68 | **0.93** | **8040.13** | 9.43 | 11.24 | 818.73 | 0.99 | **1547.76** | 7.37 | 7.70 | **158.24** |
| GNN-Explorer w/ Smoother | **0.98** | 10447.45 | 1.74 | 1.90 | 730.69 | 0.91 | 14987.09 | 18.73 | 19.70 | 2083.88 | 0.98 | 5563.40 | 8.94 | 9.22 | 789.72 |
| GraphMP w/ Smoother | 0.97 | 8232.43 | 1.27 | 1.69 | 514.19 | 0.89 | 14335.02 | 10.45 | 13.64 | 1623.25 | 0.96 | 3460.92 | 7.60 | 9.05 | 302.87 |
| BrainyMP w/ Smoother | **0.98** | 7603.48 | 1.30 | 1.47 | 362.89 | 0.90 | 10866.84 | 9.20 | 11.18 | 1012.61 | 0.99 | 1933.59 | 7.10 | 7.47 | 186.51 |
| NeuroMP w/ Smoother | **0.98** | 5967.74 | 1.25 | **1.45** | 269.26 | **0.93** | 8900.30 | 9.01 | **10.44** | 852.25 | 0.99 | 1861.49 | 7.08 | **7.40** | 185.83 |

| Methods | Kuka13 | | | | | Kuka2Arms | | | | | Kuka3Arms | | | | |
|---|---|---|---|---|---|---|---|---|---|---|---|---|---|---|---|
| | SR↑ | CK↓ | PC↓ | PCP↓ | PT↓ | SR↑ | CK↓ | PC↓ | PCP↓ | PT↓ | SR↑ | CK↓ | PC↓ | PCP↓ | PT↓ |
| RRT* | 0.68 | 2981.80 | 9.15 | 30.47 | 269.91 | 0.69 | 2810.07 | 9.69 | 29.87 | 203.75 | 0.25 | 2986.57 | 11.38 | 40.23 | 666.92 |
| BIT* | **1.00** | 2223.43 | 12.07 | 12.22 | 209.14 | **1.00** | 1559.93 | 12.05 | 12.05 | 108.99 | 0.57 | 7947.42 | 17.79 | 31.80 | 1056.73 |
| LazySP | 0.99 | 435.29 | 16.78 | 16.98 | 96.30 | 0.99 | 576.25 | 16.12 | 16.52 | 157.96 | 0.56 | 2636.75 | 21.19 | 44.63 | 844.80 |
| NEXT | 0.61 | 4868.52 | 10.35 | 48.58 | 3830.19 | 0.66 | 4637.57 | 10.26 | 48.74 | 3719.14 | 0.38 | 2141.68 | 15.39 | 37.25 | 4383.60 |
| GNN-Explorer | 0.99 | 741.34 | 15.75 | 15.89 | 104.35 | 0.99 | 574.80 | 16.60 | 16.94 | 110.91 | 0.57 | 2880.16 | 20.62 | 33.40 | 764.84 |
| GraphMP | 0.99 | 645.51 | 11.42 | 12.68 | 124.71 | 0.98 | 581.96 | 11.19 | 11.77 | 88.94 | 0.60 | 1875.29 | 16.37 | 25.68 | 412.06 |
| BrainyMP | **1.00** | 406.77 | 9.40 | 9.49 | 74.83 | 0.99 | 405.82 | 11.43 | 11.69 | 62.94 | **0.64** | 1575.30 | 16.13 | 24.75 | 406.92 |
| NeuroMP | **1.00** | 392.33 | 9.32 | 9.40 | 69.53 | 0.99 | **385.06** | 10.33 | 10.53 | 62.81 | **0.64** | **1266.69** | 16.04 | 24.66 | **314.64** |
| GNN-Explorer w/ Smoother | 0.99 | 930.81 | 9.98 | 10.03 | 143.27 | 0.99 | 804.58 | 9.86 | 9.92 | 149.48 | 0.57 | 3039.68 | 11.85 | 21.92 | 784.37 |
| GraphMP w/ Smoother | 0.99 | 700.22 | 9.22 | 9.60 | 138.07 | 0.98 | 646.40 | 9.57 | 10.00 | 99.56 | 0.60 | 1979.20 | **10.73** | 20.22 | 469.52 |
| BrainyMP w/ Smoother | **1.00** | 451.60 | 8.60 | 8.68 | 85.77 | 0.99 | 472.40 | 9.50 | 9.79 | 71.35 | **0.64** | 1671.10 | 11.06 | 19.70 | 422.95 |
| NeuroMP w/ Smoother | **1.00** | 435.26 | **8.57** | **8.65** | 79.31 | 0.99 | 433.58 | **9.15** | **9.37** | 70.98 | **0.64** | 1362.04 | 10.98 | **19.61** | 319.92 |

Table 2: Comparison of collision probability prediction between Neural Collision Checker and Perceptive Segment Selector in different environments.

| Methods | Stick3 | | | | Link8 | | | | Ur5 | | | |
|---|---|---|---|---|---|---|---|---|---|---|---|---|
| | Accuracy↑ | Recall↑ | F1↑ | Confidence↑ | Accuracy↑ | Recall↑ | F1↑ | Confidence↑ | Accuracy↑ | Recall↑ | F1↑ | Confidence↑ |
| Neural Collision Checker | 97.47 | 98.39 | 97.92 | 97.71 | 95.40 | 95.44 | 95.42 | 95.97 | 96.77 | 98.02 | 97.39 | 95.88 |
| Perceptive Segment Selector | **98.62** | **99.32** | **98.97** | **98.74** | **96.93** | **97.38** | **97.16** | **97.30** | **97.49** | 95.58 | 96.53 | **98.02** |

| Methods | Kuka13 | | | | Kuka2Arms | | | | Kuka3Arms | | | |
|---|---|---|---|---|---|---|---|---|---|---|---|---|
| | Accuracy↑ | Recall↑ | F1↑ | Confidence↑ | Accuracy↑ | Recall↑ | F1↑ | Confidence↑ | Accuracy↑ | Recall↑ | F1↑ | Confidence↑ |
| Neural Collision Checker | 91.09 | 91.26 | 91.17 | 85.44 | 91.97 | 94.92 | 93.42 | 84.20 | 89.66 | 86.54 | 88.08 | 90.81 |
| Perceptive Segment Selector | **92.89** | **94.23** | **93.55** | **94.08** | **93.88** | **95.67** | **94.76** | **95.09** | **91.11** | **88.39** | **89.73** | **92.24** |

## 5.3 Ablation Studies

We further discuss the effectiveness of five key components in NeuroMP to validate design choices, with experimental results presented in Table 3.

Table 3: Comparison of different components on NeuroMP performance.

| Methods | Stick3 | | | | Link8 | | | | Ur5 | | | |
|---|---|---|---|---|---|---|---|---|---|---|---|---|
| | SR↑ | CK↓ | PC↓ | PT↓ | SR↑ | CK↓ | PC↓ | PT↓ | SR↑ | CK↓ | PC↓ | PT↓ |
| w/o Selective Sampling | 0.96 | 5499.89 | 1.36 | 251.48 | 0.91 | 8874.08 | 11.01 | 863.53 | 0.97 | 1758.08 | 8.42 | 187.96 |
| w/o Perceptive Segment Selector | 0.97 | 5386.01 | **1.32** | 280.86 | 0.89 | 8642.14 | 10.81 | 844.62 | 0.96 | 1636.90 | **7.35** | 188.03 |
| w/o Spectral Feature | 0.97 | 5537.49 | 1.37 | 234.20 | 0.90 | 8943.50 | 11.04 | 810.24 | 0.96 | 1653.71 | 7.41 | 187.68 |
| w/o A2FL | 0.97 | 5420.61 | 1.37 | 558.58 | 0.91 | 8425.45 | 9.59 | 819.29 | 0.98 | 1652.60 | 7.74 | 172.14 |
| w/o Shortcut Retrieval | **0.98** | **4571.21** | 1.36 | **233.70** | **0.93** | **7928.49** | 9.92 | 813.78 | **0.99** | **1475.69** | 7.61 | **155.10** |
| NeuroMP | **0.98** | 5327.99 | **1.32** | 248.68 | **0.93** | 8040.13 | **9.43** | 818.73 | **0.99** | 1547.76 | 7.37 | 158.24 |

| Methods | Kuka13 | | | | Kuka2Arms | | | | Kuka3Arms | | | |
|---|---|---|---|---|---|---|---|---|---|---|---|---|
| | SR↑ | CK↓ | PC↓ | PT↓ | SR↑ | CK↓ | PC↓ | PT↓ | SR↑ | CK↓ | PC↓ | PT↓ |
| w/o Selective Sampling | 0.98 | 419.92 | 10.90 | 94.55 | 0.98 | 388.11 | 10.96 | 75.84 | 0.63 | 1267.21 | 16.11 | 320.54 |
| w/o Perceptive Segment Selector | 0.98 | 415.80 | 9.43 | 93.13 | 0.98 | 409.83 | 10.43 | 82.09 | 0.62 | 1267.89 | 16.49 | 324.66 |
| w/o Spectral Feature | 0.98 | 412.64 | 9.65 | 83.77 | 0.98 | 389.99 | 10.74 | 71.23 | 0.63 | 1265.51 | 16.47 | 316.01 |
| w/o A2FL | 0.98 | 410.46 | 9.58 | 85.87 | **0.99** | 387.77 | 10.94 | 77.29 | 0.63 | 1272.37 | 16.58 | 316.36 |
| w/o Shortcut Retrieval | **1.00** | **374.77** | 9.85 | **68.07** | **0.99** | **366.70** | 11.05 | **61.65** | **0.64** | **1256.16** | 16.88 | **313.67** |
| NeuroMP | **1.00** | 392.33 | **9.32** | 69.53 | **0.99** | 385.06 | **10.33** | 62.81 | **0.64** | 1266.69 | **16.04** | 314.64 |

**Selective Sampling.** Traditional uniform sampling during RGG construction may introduce redundant nodes, increasing search complexity and reducing planning efficiency. Selective sampling addresses this by prioritizing samples in high-value regions, reducing unnecessary exploration. Experimental results show that removing selective sampling leads to substantial increases in collision checks and planning time, indicating slower convergence and higher search overhead. In addition, the observed increase in the path cost further confirms its contribution to improving the path quality. Overall, selective sampling enhances planning efficiency and solution quality while maintaining feasibility.

**Perceptive Segment Selector** $\mathcal{N}_S$**.** The module $\mathcal{N}_S$ predicts edge-wise collision probabilities within the RGG and removes potentially dangerous edges to generate a safer RGG$'$. This process significantly reduces the search space, improving planning efficiency. Compared to the Neural Collision Checker in GraphMP, the $\mathcal{N}_S$ offers more accurate filtering, mitigating invalid path exploration and enhancing search efficiency. Ablation studies show that replacing $\mathcal{N}_S$ with Neural Collision Checker increases collision checks and planning time, suggesting that Neural Collision Checker retains many unsafe edges, thereby inflating computational overhead. In contrast, the $\mathcal{N}_S$ enables more reliable graph pruning, improving downstream search stability and performance.

**Global Alignment Heuristic.** We analyze the effect of this module by incrementally introducing its two core components.

(i) *Introducing Spectral Features.* Traditional GNNs rely on local neighborhood information, which can compromise topological consistency in weakly connected graphs, hindering global path coherence. The module $\mathcal{N}_H$ addresses this by incorporating spectral features to enforce global topological constraints and enhance structural representation. Ablation studies show that removing spectral features results in increased path costs, higher collision checks, and reduced success rates in several environments. These findings demonstrate the importance of spectral features in preserving path continuity, reducing search overhead, and improving planning reliability.

(ii) *Alignment Dual-Flow Learning (A2FL).* GNN-based methods are often limited by sparse connectivity, making it challenging for some critical nodes to obtain effective heuristic values. To address this, A2FL incorporates a completed channel to assist the weak channel during training, enhancing information propagation in sparse graphs. Consequently, the $\mathcal{N}_H$ can provide more accurate heuristic estimations in weakly connected graphs. Experimental results indicate that NeuroMP trained without A2FL exhibits increased collision checks, higher path curvature, and longer planning times across various environments. These findings demonstrate that A2FL improves the estimation capability of $\mathcal{N}_H$, reducing redundant exploration and improving path quality.

**Shortcut Retrieval.** To reduce unnecessary detours, NeuroMP employs the Shortcut Retrieval step to remove redundant nodes from the searched path. Ablation experiments indicate that although this step slightly increases collision checks and planning time, it significantly reduces path redundancy and enhances path quality. The time complexity of this step is $O(T^2)$, where $T$ represents the number of nodes in the searched path. Since the A* block typically generates short paths, this step effectively reduces path costs with minimal computational overhead.

## 5.4 Parameters Discussion

We empirically set the range of the maximum sample number, $k$-values, and the bias sampling probability $\beta$ and collision probability threshold $\alpha$ to $[200, 700]$, $[20, 30]$, $[0.2, 0.4]$, and $[0, 0.1]$, respectively. A more detailed results and analysis of these parameter settings are in Appendix C.

## 5.5 Further Performance Comparison of Weakly Connected Graphs

In weakly connected graphs with a collision threshold of $0.9$, we present the retained edge ratio and planning performance of NeuroMP, BrainyMP and GraphMP. Results indicate that NeuroMP demonstrates enhanced stability in sparse graphs and improves planning performance. Detailed results and analyses are provided in the Appendix D.

## 6 Conclusion

Inspired by the two-stage Perception-Decision model, this paper presents NeuroMP, a two-stage brain-inspired motion planning framework to enhance planning performance in high-dimensional spaces. Across various motion planning tasks, NeuroMP demonstrates superior planning efficiency and path quality, highlighting the potential of brain-inspired methods to improve robotic decision making. Future work will focus on extending the model to dynamic, large-scale, and real-world scenarios while further enhancing computational efficiency.

## Acknowledgments

This work is supported by the State Key Program of National Natural Science of China (Grant No. 62236001), the Natural Science Foundation of Beijing, China (Grant No. L247011), the Major Research Plan of the National Natural Science Foundation of China (Grant No. 92470125), and Hebei Natural Science Foundation (Grant No. F2024105033). It is also supported by the MoE Key Laboratory of Brain-inspired Intelligent Perception and Cognition, University of Science and Technology of China (Grant No. 2421001) and Key Laboratory of Machine Intelligence and System Control, Ministry of Education (Grant No. MISC-202403).

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

# A    Methodology

## A.1    Differentiable A*

Differentiable A* [18] redefines the traditional A* algorithm [3], enabling end-to-end differentiable training. The open list and closed list are represented by binary vectors $\mathbf{o} \in [0, 1]^{|\mathcal{V}|}$ and $\mathbf{c} \in [0, 1]^{|\mathcal{V}|}$, respectively. The candidate node $\mathbf{v}^*$ with the lowest path cost is then selected based on the following calculations.

$$\mathbf{v}^* = \mathcal{I}_{\max}(\frac{\exp(-(\mathbf{G} + \mathbf{H})/\lambda) \odot \mathbf{o}}{\langle \exp(-(\mathbf{G} + \mathbf{H})/\lambda), \mathbf{o} \rangle}), \tag{9}$$

$$\mathbf{o} = \mathbf{o} - \mathbf{v}^*, \mathbf{c} = \mathbf{c} - \mathbf{v}^*, \tag{10}$$

where $\mathbf{G}$ is the cumulative cost, $\mathbf{H}$ is the heuristic value estimated by the neural network, and $\lambda$ is a preset parameter. $\mathcal{I}$ is the function that returns a one-hot vector, and the element-wise product $\odot$ is used to mask the nodes.

During the node expansion process, the neighbors of candidate nodes can be computed as $\mathbf{v_{nbr}} = \mathbf{A}\mathbf{v}^* \odot (\mathbb{1} - \mathbf{c})$, where $\mathbf{v_{nbr}}$ is a binary vector, with the entries corresponding to neighboring nodes marked as 1, $\mathbf{A} \in [0, 1]^{|\mathcal{V}| \times |\mathcal{V}|}$ is the adjacency matrix, and $\mathbb{1}$ is a vector of all ones. Subsequently, $\mathbf{G}$ and $\mathbf{o}$ are iteratively updated as the node expansion process progresses,

$$\mathbf{G}' = \mathbf{G} \odot \mathbf{v}^* + \mathbf{W}\mathbf{v}^*, \tag{11}$$

$$\Psi = ((\mathbb{1} - \mathbf{o}) + \mathbf{o} \odot (\mathbf{G} > \mathbf{G}')) \odot \mathbf{v_{nbr}},$$
$$\mathbf{G} = \mathbf{G} \odot (\mathbb{1} - \Psi) + \mathbf{G}' \odot \Psi, \tag{12}$$

$$\mathbf{o} = \mathbf{o} + (1 - \mathbf{o})\mathbf{v_{nbr}}, \tag{13}$$

where $\mathbf{G}'$ denotes the accumulated cost of the nodes along the path, including the currently selected nodes, $\mathbf{W} \in \mathbb{R}^{|\mathcal{V}| \times |\mathcal{V}|}$ is the weighted adjacency matrix, and $\mathbf{W}\mathbf{v}^*$ computes the distance from $\mathbf{v}^*$ to each of its neighboring vertices. The accumulated costs of the neighboring nodes are then updated according to Eq.12. Subsequently, $\mathcal{O}$ is expanded by incorporating the newly explored neighbors.

## A.2    Optimization in Online Planning

When NeuroMP performs online planning for a new problem, the trained modules $\mathcal{N}_S$, $\mathcal{N}_H$, and A* are integrated to perform the graph search. To further improve path quality and efficiency, two optimization steps are incorporated during the inference procedure of online planning.

**Selective Sampling.** In traditional graph construction, uniform sampling methods often ignore the environmental structure, leading to purposeless sampling. To address this, we employ a selective sampling approach, where a hyperparameter $\beta \in [0, 1]$ controls the balance between biased and uniform sampling, optimizing exploration efficiency while preserving probabilistic integrity. Specifically, the robot samples nodes from a guiding region with probability $\beta$, and uniformly samples from the entire configuration space with probability $1 - \beta$. The guiding region is the circular area with the start and goal as its diameter, reducing redundant sampling.

**Shortcut Retrieval.** Although the A* algorithm completes the pathfinding, the resulting path may have detours. Shortcut retrieval [9] is applied to improve path quality. A check window is defined to identify collision-free shortcut edges. The window length is set to 2. If found, these two nodes are directly connected, and then redundant intermediate nodes are removed, eliminating unnecessary detours.

# B    Detailed Overall Performance Analysis

Table 1 presents a comparative result of NeuroMP and other baselines across six environments. Across all environments, the maximum sampling number is limited to 1000 for all methods, except for RRT* and NEXT in the Link8 environment, where it is increased to 2000 due to their low success rates (approximately 0.2). By leveraging selective sampling, accurate collision prediction, and optimized A* search, NeuroMP achieves a high success rate while significantly reducing collision checks and planning time, maintaining competitive path quality.

**Success Rate.** NeuroMP consistently achieves optimal or near-optimal success rates across all environments, demonstrating strong competitiveness with state-of-the-art (SOTA) planners. In contrast, RRT* exhibits low success rates, reaching only 0.41 and 0.39 in the Link8 and Ur5 environments, highlighting the instability of classical planners in high-dimensional settings. BIT* and LazySP, benefiting from manually designed heuristics, achieve high success rates in most scenarios. While the CNN-based planner NEXT performs adequately in simpler 3D environments, its success rate drops sharply as dimensionality increases, attaining only 0.41 and 0.37 in Link8 and Ur5, respectively. By comparison, GNN-based planners exhibit consistently superior success rates across all environments.

**Collision Check.** The collision check is a critical factor influencing planning time and computational efficiency, with fewer checks indicating more optimized search spaces. NeuroMP achieves significantly fewer collision checks than SOTA methods across all environments. In addition, NeuroMP significantly reduces collision checks compared to GraphMP and GNN-Explorer in all environments, requiring only 71%, 61%, 57%, 61%, 66%, and 68% of the collision checks performed by GraphMP. This reduction highlights the effectiveness of its precise edge selection and efficient heuristic search. While LazySP benefits from a lazy collision check strategy, NeuroMP achieves further improvements, reducing collision checks to 69%, 77%, 57%, 90%, 67%, and 48% of those by LazySP. Even in the Kuka2Arms and Kuka3Arms environments, NeuroMP maintains low collision check counts of 385.06 and 1266.69, respectively, outperforming all baselines. Furthermore, although NEXT and RRT* show lower path costs, their extremely low success rates indicate poor reliability, as they can only solve relatively simple cases.

**Planning Time.** A common concern with learning-based methods is their high computational cost due to frequent neural network evaluations, such as in NEXT. However, GNN-based planning methods mitigate this issue by operating on sampled RGGs, substantially reducing planning time. Compared to SOTA planners, NeuroMP achieves competitive or faster planning across environments. Although classical planners like RRT* exhibit lower planning time due to their simplicity, they suffer from low reliability. Compared to RRT*, NeuroMP achieves comparable planning time in Stick, while providing $0.53\times$, $0.31\times$, $2.88\times$, $2.24\times$, and $1.12\times$ speedups in the Link8-Kuka3Arms environments.

**Path cost and Path Cost with Penalty.** NeuroMP w/ Smoother achieves the lowest path cost from the Kuka13 to the Kuka3Arms environment. Although RRT* and NEXT occasionally achieve lower path costs in some environments, their low success rates result in significant penalty costs due to frequent failures. The path cost with penalty confirms that NeuroMP maintains superior path quality even under high-success-rate conditions. Furthermore, NeuroMP outperforms the SOTA planner in environments with high obstruction density (Stick3) and high-dimensional spaces (Kuka3Arms), highlighting its robustness in complex conditions.

## C  Parameters Discussion

### C.1  Maximum Sampling Number

Fig. 5 illustrates the impact of varying maximum sampling numbers on NeuroMP's performance. In GNN-based planners, the number of sampled nodes directly influences graph density, connectivity, and computational complexity. Specifically, increasing the number of sampled nodes results in a denser graph, covering more feasible areas and reducing failures caused by insufficient sampling. However, once the sampling density reaches the coverage threshold, additional nodes contribute less to finding new paths, causing the success rate to plateau. Similarly, denser graphs may contain more optimal paths, leading to a slight reduction in path cost. However, collision checks and planning time increase significantly as more nodes and edges expand the search space, thereby increasing collision checks and computational costs.

### C.2  $k$ Value

We assume sufficient sample points are available, with the maximum number of samples set within the range $[300, 700]$. Fig. 6 illustrates the impact of varying $k$-values on NeuroMP's performance. As the $k$ value increases, each node connects to more neighboring nodes, constructing a denser edge structure that enhances overall graph connectivity, increasing the success rate. However, when the $k$-value exceeds the actual connectivity requirement for the current environment, the success rate improvement

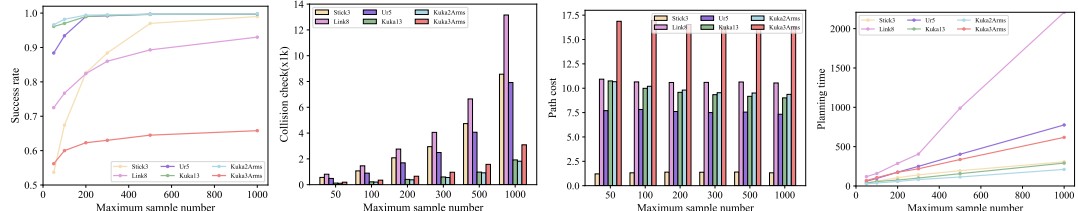

Figure 5: Maximum sampling number. (a) Success rate, (b) Collision check, (c) Path cost, (d) Planning time.

plateaus or slightly decreases. A denser edge structure also increases the likelihood of finding shorter paths, which reduces path cost but slightly increases collision checks. The planning time initially decreases, benefiting from improved heuristic search efficiency due to higher graph connectivity, but excessive edges introduce redundancy, leading to longer computation times. Therefore, selecting an appropriate $k$ value based on environmental characteristics is essential. In our experimental setup, the optimal threshold range is set to $[20, 30]$.

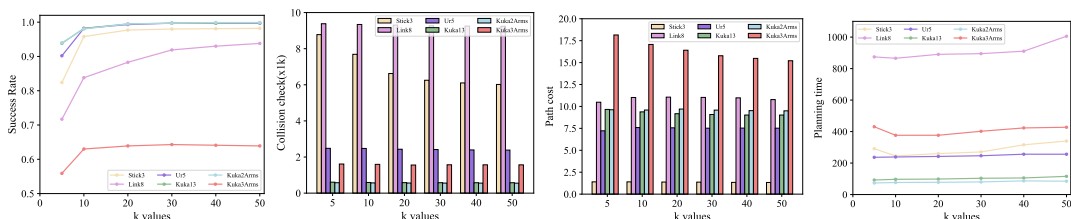

Figure 6: The $k$ value of K-NN. (a) Success rate, (b) Collision check, (c) Path cost, (d) Planning time.

### C.3 The Bias Sampling Probability $\beta$

Fig. 7 presents the performance of NeuroMP under different $\beta$ values without applying shortcut retrieval. The success rate initially increases and then decreases as $\beta$ grows, since a higher sampling density in the connection region between start and goal improves the likelihood of discovering feasible paths. However, excessive concentration in this region reduces exploration diversity, making it harder to circumvent dense obstacles, thus increasing failure cases. Collision checks generally decline with increasing $\beta$, as samples in the connection region are more likely to form coherent path segments, reducing redundant edge validation. Moreover, high-density sampling in this area expedites the discovery of near-straight-line paths, potentially lowering path costs. Nonetheless, overly concentrated nodes can lead to local detours around obstacles, increasing overall path cost. Planning time also varies with environment characteristics: in low-obstacle settings (e.g., Kuka13), concentrated sampling reduces RGG size and accelerates planning; in high-obstacle settings (e.g., Stick3), it may trigger more frequent backtracking in the A* search.

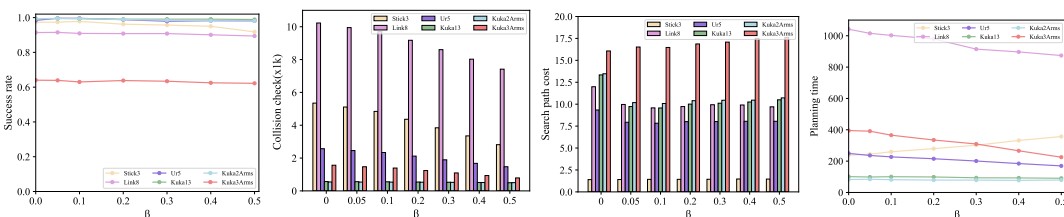

Figure 7: Varying bias sampling probability $\beta$. (a) Success rate, (b) Collision check, (c) Path cost, (d) Planning time.

## C.4 Collision Probability Threshold $\alpha$

Fig. 8 illustrates NeuroMP's performance under varying $\alpha$ values, with the maximum sampling number set within $[300, 700]$ and shortcut retrieval disabled. As $\alpha$ increases, more high-risk edges are removed, reducing path interruptions during A* search and improving success rates. However, when $\alpha > 0.5$, excessive filtering degrades graph connectivity, severing critical connections and increasing failure rates due to the inability to bypass dense obstacles. Without filtering, infeasible edges can obstruct viable paths; conversely, over-filtering reduces the likelihood of finding collision-free paths. Path cost rises with $\alpha$ as detours become more frequent, while planning time decreases due to reduced traversed edges. These results indicate that $\alpha$ must balance safety (filtering risky edges) and connectivity (preserving feasible paths), with optimal performance observed in the range $[0.2, 0.4]$.

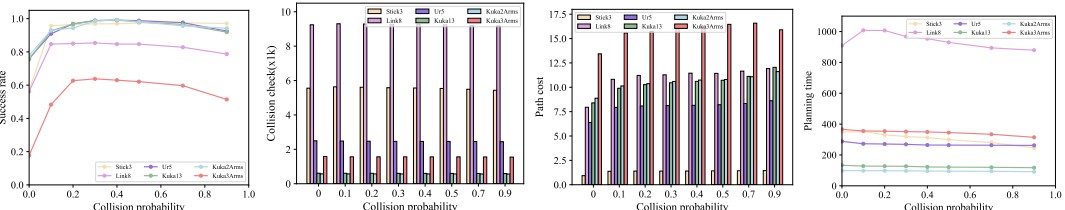

Figure 8: Varying collision probability threshold $\alpha$. (a) Success rate, (b) Collision check, (c) Path cost, (d) Planning time.

## D Further Performance Comparison of Weakly Connected Graphs

In weakly connected graphs with a collision threshold of 0.9, Table 4 presents the retained edge ratio and planning performance of NeuroMP, BrainyMP, and GraphMP. When $\alpha = 0.9$, most high-risk edges are removed, allowing NeuroMP to surpass GraphMP in planning efficiency and path quality, demonstrating a significant advantage in success rates. When selector performance is comparable, NeuroMP produces more informative heuristics than BrainyMP, yielding significantly faster searches. In more complex settings (e.g., Kuka13 and Kuka2Arms), NeuroMP's selector more accurately removes risky edges and demonstrates a significant advantage in success rates. In denser graphs (with more retained edges), NeuroMP achieves slightly higher efficiency while still producing low-cost paths. These results indicate that NeuroMP demonstrates enhanced stability in sparse graphs and improves planning performance.

Table 4: Performance comparison between NeuroMP and GraphMP, BrainyMP with $\alpha = 0.9$.

| Methods | Stick3 | | | | | | Link8 | | | | | | Ur5 | | | | | |
|---|---|---|---|---|---|---|---|---|---|---|---|---|---|---|---|---|---|---|
| | ratio | SR↑ | CK↓ | PC↓ | PCP↓ | PT↓ | ratio | SR↑ | CK↓ | PC↓ | PCP↓ | PT↓ | ratio | SR↑ | CK↓ | PC↓ | PCP↓ | PT↓ |
| GraphMP | 0.68 | 0.96 | 7098.45 | **1.40** | 1.87 | 254.03 | 0.41 | 0.72 | 10365.27 | 9.93 | 17.00 | 946.73 | 0.22 | 0.83 | 2746.88 | 8.98 | 14.97 | 265.22 |
| BrainyMP | 0.69 | 0.96 | 7300.75 | 1.41 | 2.12 | 270.86 | 0.41 | 0.77 | 9886.37 | **8.85** | **14.86** | 930.23 | 0.22 | 0.90 | 2639.14 | 8.05 | 11.05 | 241.10 |
| NeuroMP | 0.63 | **0.97** | **6664.90** | **1.40** | **1.74** | 245.35 | 0.36 | **0.79** | 9275.09 | 10.07 | 15.36 | **879.28** | 0.24 | **0.93** | **2444.38** | **7.87** | **10.91** | 262.14 |
| Methods | Kuka13 | | | | | | Kuka2Arms | | | | | | Kuka3Arms | | | | | |
| | ratio | SR↑ | CK↓ | PC↓ | PCP↓ | PT↓ | ratio | SR↑ | CK↓ | PC↓ | PCP↓ | PT↓ | ratio | SR↑ | CK↓ | PC↓ | PCP↓ | PT↓ |
| GraphMP | 0.26 | 0.55 | 613.38 | 10.42 | 21.42 | 121.48 | 0.31 | 0.62 | 574.92 | 9.98 | 20.00 | **91.49** | 0.22 | 0.44 | 1571.26 | **15.67** | 28.66 | 333.52 |
| BrainyMP | 0.26 | 0.52 | **543.24** | 10.29 | 24.29 | **106.85** | 0.31 | 0.64 | **570.87** | **9.71** | 19.29 | 85.03 | 0.22 | 0.46 | 1571.19 | 16.47 | 29.21 | 358.79 |
| NeuroMP | 0.46 | **0.91** | 624.66 | **10.02** | **12.75** | 117.55 | 0.55 | **0.94** | 589.63 | 10.13 | **11.69** | 95.73 | 0.21 | **0.52** | **1568.66** | 15.90 | **27.61** | **314.44** |

## E Discussion on Map Size and Obstacle Density

### E.1 Different Map Sizes

Under fixed robot shape (e.g., point robot) and obstacle density, we evaluate the performance of NeuroMP, BrainyMP, and GraphMP across various map sizes, as shown in Fig. 9, with the maximum number of samples limited to 1000. As the map size increases, all methods exhibit reduced path quality and planning efficiency, but the degradation is more gradual for NeuroMP. Specifically, while overall success rates decline, NeuroMP sustains a higher success rate on the $50 \times 50$ map, whereas GraphMP shows a marked drop, highlighting NeuroMP's superior robustness in large-scale environments. The increased map size introduces more nodes and edges, elevating collision

checks and computational costs, thereby reducing search efficiency. Nevertheless, compared to GraphMP and BrainyMP, NeuroMP performs more efficient path searches and substantially reduces unnecessary collision checks by approximately 60% and 40% on the $50 \times 50$ map. Although path costs increase due to longer step sizes, NeuroMP consistently generates lower-cost paths. While it exhibits faster planning on smaller maps, planning time increases notably in larger maps. Future work will aim to further optimize computational efficiency in large-scale environments to enhance practical applicability.

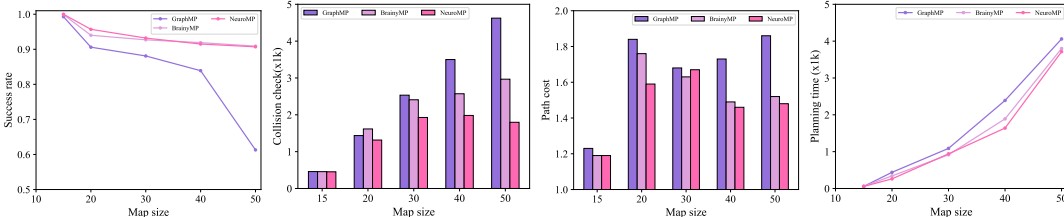

Figure 9: Performance comparison of NeuroMP, BrainyMP and GraphMP under different map sizes environments. (a) Success rate, (b) Collision check, (c) Path cost, (d) Planning time.

Additionally, Table 5 compares the collision prediction performance of Perceptive Segment Selector $\mathcal{N}_S$ in NeuroMP and Neural Collision Checker in GraphMP and BrainyMP across different map sizes. The module $\mathcal{N}_S$ consistently outperforms Neural Collision Checker, particularly on smaller maps (from 15 to 30), with significantly higher accuracy, recall, F1 score, and confidence. Although all methods experience a decline in prediction quality on larger maps, NeuroMP maintains clear advantages. Moreover, NeuroMP achieves consistently higher confidence levels, indicating more stable and reliable predictions, thereby enabling more accurate collision detection across varying environmental scales.

Table 5: Comparison of collision probability prediction between Neural Collision Checker and Perceptive Segment Selector in different map size environments.

| Map Size | Methods | Accuracy↑ | Recall↑ | F1↑ | Confidence↑ |
|---|---|---|---|---|---|
| 15 | Neural Collision Checker | 99.23 | 99.37 | 99.30 | 97.72 |
| | Perceptive Segment Selector | **99.43** | **99.71** | **99.57** | **99.45** |
| 20 | Neural Collision Checker | 97.29 | 97.96 | 97.63 | 97.28 |
| | Perceptive Segment Selector | **99.32** | **99.48** | **99.40** | **99.36** |
| 30 | Neural Collision Checker | 93.45 | 91.01 | 92.22 | 94.09 |
| | Perceptive Segment Selector | **94.33** | **92.85** | **93.58** | **94.73** |
| 40 | Neural Collision Checker | 85.90 | 73.00 | 78.93 | 86.88 |
| | Perceptive Segment Selector | **86.69** | **74.99** | **80.42** | **87.72** |
| 50 | Neural Collision Checker | 85.33 | 61.33 | 71.37 | 85.90 |
| | Perceptive Segment Selector | **85.46** | **62.53** | **72.22** | **85.95** |

## E.2 Varying Obstacle Densities

We construct three obstacle density levels in a $15 \times 15$ maze environment based on point robots. The obstacle densities for the easy, normal, and hard environments are set to 26%-36%, 36%-47%, and 47%-60%, respectively. Fig. 10 illustrates the performance of NeuroMP, BrainyMP, and GraphMP under these varying densities, with the maximum number of samples limited to 1000. Although success rates slightly decline as obstacle density increases, NeuroMP consistently achieves 100%, 99.95%, and 99.90% success rates in the respective scenarios. As complexity increases, all methods require more collision checks and longer planning times. However, NeuroMP outperforms both GraphMP and BrainyMP across all settings, requiring only about one-third the number of collision checks compared to GraphMP. This improvement is attributed to the more accurate filtering of high-risk edges by NeuroMP, which effectively reduces redundant checks. Moreover, while GraphMP exhibits a notable increase in path cost under high-density conditions, NeuroMP maintains more stable and reliable path quality.

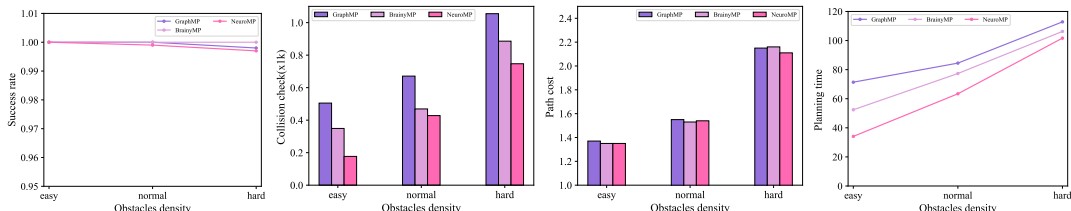

Figure 10: Performance comparison of NeuroMP, BrainyMP and GraphMP under varying obstacle densities. (a) Success rate, (b) Collision check, (c) Path cost, (d) Planning time.

Furthermore, Table 6 compares the module $\mathcal{N}_S$ and Neural Collision Checker in collision prediction across varying obstacle densities. The module $\mathcal{N}_S$ consistently outperforms Neural Collision Checker in accuracy and confidence across all difficulty levels, demonstrating more stable and reliable predictions that enhance collision checking in complex environments.

Table 6: Comparison of collision probability prediction between Neural Collision Checker and Perceptive Segment Selector in environments with different obstacle densities.

| Obstacle Density | Methods | Accuracy↑ | Recall↑ | F1↑ | Confidence↑ |
|---|---|---|---|---|---|
| Easy | Neural Collision Checker | 99.37 | 99.59 | 99.48 | 99.22 |
| | Perceptive Segment Selector | **99.74** | **99.87** | **99.80** | **99.77** |
| Normal | Neural Collision Checker | 99.05 | 99.19 | 99.12 | 97.13 |
| | Perceptive Segment Selector | **99.30** | **99.61** | **99.46** | **99.38** |
| Hard | Neural Collision Checker | 99.02 | 98.20 | 99.21 | 99.20 |
| | Perceptive Segment Selector | **99.57** | **99.73** | **99.65** | **99.60** |

## F    Discussion of the U-shaped environment

The U-shaped environment is a canonical challenge in motion planning, so U-shape tests were added to further validate our method. In a 15×15 2D workspace, U-shaped obstacles are generated to simulate narrow passages between shelves. The starting point is positioned near the bottom inside the U-shaped obstacle, while the goal is placed outside the U-shaped obstacle. The tests are conducted with stick and link8 robots on 500 U-shaped map instances. The results for both robots in the U-shaped environment are shown in the Table 7. Our method shows significant advantages in reasoning efficiency, success rate, and path quality.

Table 7: Comparison of all methods in the U-shaped environment.

| Methods | Stick3 | | | | | Link8 | | | | |
|---|---|---|---|---|---|---|---|---|---|---|
| | SR↑ | CK↓ | PC↓ | PCP↓ | PT↓ | SR↑ | CK↓ | PC↓ | PCP↓ | PT↓ |
| RRT* | 0.68 | 7290.00 | **1.44** | 6.81 | 84.34 | 0.68 | 6187.61 | **6.21** | 16.75 | 450.11 |
| BIT* | **1.00** | 4704.71 | 1.62 | 1.62 | 60.04 | **0.95** | 13545.35 | 10.31 | 12.29 | 582.36 |
| LazySP | 0.99 | 1856.48 | 1.73 | 1.78 | 31.61 | 0.92 | 6829.40 | 11.36 | 13.36 | 927.80 |
| NEXT | 0.99 | 2051.83 | 1.42 | 1.68 | 115.14 | 0.63 | 9656.06 | 7.65 | 19.36 | 4838.95 |
| GNN-Explorer | **1.00** | 1419.12 | 2.19 | 1.78 | 33.77 | 0.93 | 7926.33 | 12.76 | 14.10 | 681.33 |
| GraphMP | 0.99 | 1791.17 | 1.68 | 1.77 | 40.08 | 0.93 | 8085.31 | 8.23 | 9.94 | 556.76 |
| BrainyMP | 0.99 | 1276.75 | 1.48 | 1.60 | 27.03 | **0.95** | 6923.27 | 8.03 | 9.49 | 367.59 |
| NeuroMP | 0.99 | **1132.01** | 1.46 | **1.55** | **24.72** | **0.95** | **5848.10** | 7.71 | **8.90** | **287.44** |

## G    Case Study in a Real-World Setting

Real-world scenarios or simulations that mimic real-world conditions can further validate the effectiveness of our method. We select the City/Street Map (CSM) Dataset as the real-world scenario

for validation. Sturtevant et al.[46] used drones to collect 30 real city maps with marked obstacles represented as binary images. Based on this, we randomly generate 2000 training maps and 1000 test instances from the 30 maps. For each map, the start and goal are randomly generated in the free space. These maps present significantly higher complexity than synthetic environments: irregular obstacle shapes, narrow alleyways, and dense urban structures that challenge traditional planners.

Table 8: Comparison of the planning performance of all methods on the CSM dataset.

| Methods | SR↑ | CK↓ | PC↓ | PCP↓ | PT↓ |
|---|---|---|---|---|---|
| RRT* | 0.82 | 1609.79 | 0.97 | 2.61 | 112.99 |
| BIT* | 0.86 | 1262.28 | 0.94 | 2.24 | 363.39 |
| LazySP | 0.85 | 662.99 | 0.98 | 2.31 | 206.71 |
| NEXT | 0.79 | 6552.28 | 0.90 | 2.78 | 2590.84 |
| GNN-Explorer | 0.85 | 805.70 | 1.04 | 2.38 | 362.55 |
| GraphMP | 0.86 | 1045.66 | 0.90 | 2.06 | 151.27 |
| BrainyMP | 0.87 | 692.84 | 0.90 | 1.94 | 125.39 |
| NeuroMP | **0.88** | **526.42** | 0.88 | 1.86 | **102.78** |
| GNN-Explorer w/ Smoother | 0.85 | 951.39 | 0.95 | 2.00 | 373.34 |
| GraphMP w/ Smoother | 0.86 | 1064.91 | 0.89 | 1.91 | 153.96 |
| BrainyMP w/ Smoother | 0.87 | 714.79 | 0.89 | 1.80 | 127.77 |
| NeuroMP w/ Smoother | **0.88** | 545.07 | **0.87** | **1.74** | 105.02 |

The results on the CSM dataset are demonstrated in Table 8. In real-world environments with fine-grained and densely distributed obstacles, robots typically require more collision checks and longer planning times. NeuroMP achieves the highest success rate (0.88) while requiring 67% fewer collision checks than RRT* and 87% faster planning than BIT*. Due to encoding the entire workspace, NEXT incurs substantially higher computational overhead compared to other methods. In contrast, our method not only achieves competitive success rates but also the fewest collision checks and the shortest planning time across all baselines. These results demonstrate superior real-world generalization of GNN-based approaches, particularly ours, offering excellent stability in complex urban environments.

# H    Limitations

Despite its strong performance, NeuroMP has the following limitations.

**Sensitivity to Perception Quality.** The Perceptive Segment Selector in NeuroMP relies on identifying and reasoning about key features of the environment for accurate collision prediction. In scenarios with significant perception noise or blurred obstacle boundaries, errors in perception may lead to inaccurate graph construction, thereby affecting the overall performance of path planning.

**Lack of Asymptotical Optimality Guarantees.** Searching on an RGG cannot guarantee inclusion of the true optimal path, thereby limiting theoretical optimality. Although our selective sampling strategy mitigates this issue to some degree, it cannot ensure optimal-path coverage. In the future, we can (i) design enhanced graph-construction strategies that leverage prior knowledge to guide sampling, reduce randomness, and increase the probability that optimal-path structures are represented; (ii) explore environment-aware dynamic pruning, which can adaptively refine the RGG online to improve coverage of critical connections and ensure robust path representation.

**Generalization Not Fully Evaluated.** Current experiments focus on maze planning and robotic arm manipulation. Systematic evaluations across broader task domains are still lacking, especially in dynamic environments or real-world scenarios. In the future, the experimental scope can be expanded to include dynamic scenarios with moving obstacles and time-varying environmental parameters, enabling assessment under nonstationary conditions. In addition, studies will be conducted in realistic settings (e.g., indoor navigation and autonomous driving), using real data to quantify performance. These extended evaluations are expected to inform architectural refinements and improve robustness and generalization in dynamic and real-world environments.

