# OpenReview forum: "Learning to Plan Like the Human Brain via Visuospatial Perception and Semantic-Episodic Synergistic Decision-Making"
_NeurIPS.cc/2025/Conference — NeurIPS 2025 poster_

### Official Review · Reviewer_snn6 · 2025-06-12

**Clarity:** 3
**Significance:** 3
**Originality:** 3
**Rating:** 4
**Confidence:** 3

**Summary:**

In this paper, the authors proposed a new two-staged planning method, NeuroMP, motivated by the multi-level planning functionalities in the human brain: perception and decision-making. NeuroMP mimics these functionalities with a Perceptive Segment Selector and Global Alignment Heuristic. The Perceptive Segment Selector encodes local obstacle information to predict collision probabilities and filter unsafe edges, producing a safe Random Geometric Graph (RGG). In the second stage, the Global Alignment Heuristic incorporates both spatial features (e.g., distance to goal) and spectral features (e.g., global graph topology through eigendecomposition) to compute heuristic values $H$ for more efficient A* search. NeuroMP demonstrates on-par or better success rates than previous methods across diverse motion planning tasks, achieving lower path costs through improved graph construction while reducing planning time via better heuristic guidance.

**Questions:**

- Random Geometric Graphs (RGGs) are mentioned in line 36 before being properly introduced. Please define this upon first use to improve readability.
- The Perceptive Segment Selector uses $T$ iterations of obstacle encoding. Did you conduct ablation studies on the number of iterations T?
- While NeuroMP demonstrates clear efficiency gains (reduced collision checks, faster planning), the improvements in fundamental planning capabilities (success rate, path quality) appear marginal. Can you identify specific scenarios where the two-stage perception-decision architecture provides qualitative advantages beyond efficiency?
- Can you update the results with the error bars to conduct the statistical significance tests?

**Ethical Concerns:**

["NO or VERY MINOR ethics concerns only"]

**Final Justification:**

The authors have satisfactorily addressed my initial concerns, including those regarding the statistical analysis. More importantly, the critical issue I raised concerning the marginal performance improvement has been resolved through their additional discussion. Therefore, I have raised my score to recommend acceptance.

**Limitations:**

yes.

**Paper Formatting Concerns:**

I didn't notice any major formatting issues in their paper.

**Quality:**

3

**Strengths And Weaknesses:**

## Strengths

**Neuroscience Inspired Architecture.** The primary strength of this work lies in its novel and principled approach to motion planning, drawing direct inspiration from the two-stage Perception-Decision model of the human brain. This cognitive science foundation provides a strong and elegant motivation for the architectural choices , particularly the separation of the Perceptive Segment Selector (mimicking visuospatial perception ) and the Global Alignment Heuristic (mimicking semantic-episodic synergy ). This is a commendable effort to bridge the gap between neuroscience and robotic planning.

**High Clarity of Presentation.** The paper is well-written and presented. The methodology is explained with good clarity, and Figures 2 and 3, in particular, provide an intuitive and comprehensive overview of the proposed NeuroMP framework and its components. This high level of detail significantly aids in understanding the complex interactions within the model and contributes positively to the potential reproducibility of the work.

**Thorough Empirical Evaluation.** The empirical evaluation is both thorough and convincing. The authors test their method across six challenging motion planning tasks, including high-DoF robotic arms , and use a comprehensive set of metrics (SR, CK, PC, PT, PCP) to provide a multi-faceted view of performance. Furthermore, the detailed ablation studies in Section 5.3 and Appendix C rigorously validate the contribution of each proposed component (e.g., Perceptive Segment Selector, Spectral Features, A2FL) and additional experimental results in Appendix E and F provide the evidences for enhanced stability and superior performances in large-scale environments. The separate analysis of collision prediction performance (Table 2) further strengthens the claims regarding the superiority of the Perceptive Segment Selector.

## Weaknesses

**Marginal Improvement in Success Rate and Path Cost on Saturated Benchmarks.** While the proposed method demonstrates clear improvements in planning efficiency (fewer collision checks, less plan time), its improvement in success rate (SR) and Path Cost (PC) over the SOTA baseline (BrainyMP) is marginal. This is likely because the SR on these benchmarks is already saturated near 100% for the best-performing methods, leaving little room for improvement. However, based on the provided results, the primary contribution of NeuroMP appears to be enhancing the efficiency, rather than fundamentally increasing the solvability of problems and the quality of the plans.

**Lack of Statistical Significant Analysis.** The paper does not report error bars or conduct statistical significance tests for its main results in Table 1. For example, the reported improvements in Path Cost and Planning Time over BrainyMP are often small (e.g., in Ur5, PC is 7.37 vs 7.39; in Kuka2Arms, PT is 62.81s vs 62.94s). Without statistical validation (e.g., running experiments with multiple random seeds and reporting mean and standard deviation), it is difficult to conclude whether these small gains are statistically significant or simply due to random experimental variance. This makes it challenging to robustly assess the true margin of improvement.

---

> ### Author Rebuttal · Authors · 2025-07-30
>
> Thank you very much for your positive comment regarding our work as "High Clarity of Presentation" and "Thorough Empirical Evaluation". We also greatly appreciate your insightful comments pointing out the weaknesses in our work. Following your suggestions, we have provided detailed point-by-point responses below. We hope our replies will adequately address your concerns.
>
> **W1. Marginal improvements in success rate and path cost on saturated benchmarks**
>
> We appreciate your detailed analysis and comprehensive evaluation of our saturated benchmark results. NeuroMP’s improvements in success rate (SR) and path cost (PC) over BrainyMP are limited, with most benefits stemming from planning efficiency improvements. We fully agree that when success rates approach 100%, the margin for further improvement is inherently narrow. These benchmarks have been extensively studied, and existing advanced algorithms already exploit much of their planning potential, pushing performance toward theoretical limits. Nonetheless, we emphasize that reductions in planning time and collision checks remain highly valuable in practice. In real‑time applications (such as autonomous driving or emergency obstacle avoidance), faster planning directly enhances responsiveness and stability. Additionally, fewer collision checks decrease the computational load, making the method more suitable for resource-constrained platforms. Next, we discuss two key areas:
>
> 1. Generalization of real-world environments
>
> Real-world scenarios or simulations that mimic real-world conditions can further validate the effectiveness of our method. Thus, we select the City/Street Map (CSM) Dataset as the real-world scenario for validation. Our approach achieves competitive success rates, requiring the fewest collision checks and the shortest planning times. Due to space limitations, detailed experimental results and analysis can be found in the response to Reviewer 7bRM under "W2. Generalizations are Not Fully Evaluated."
>
> 2. Future work
>
> We recognize the need for further improvements in solvability and plan quality for NeuroMP. Our plans include: (i) Challenging Test Scenarios: Evaluating NeuroMP in dynamic and multi-objective conflict environments to enhance solvability and planning quality. (ii) Enhancing plan quality. Refining graph construction with advanced cost metrics, incorporating environmental risk and execution difficulty, and exploring reinforcement learning to optimize path-search strategies. (iii) Comprehensive comparisons. We will systematically compare NeuroMP and BrainyMP across a wider range of unsaturated benchmarks to more fully characterize each method’s strengths and areas for improvement.
>
> **W2 & Q3. Lack of statistical significant analysis**
>
> We fully agree that omitting error bars and significance tests can undermine confidence in the reported findings, especially when differences in metrics such as path cost and planning time are subtle. We computed the average and standard deviation of four metrics (SR, CK, PC, PT) for all methods using four different random seeds. The results are presented in Tables 1-4, where the results for GNN-based methods exclude the GNN-smoother step. In most environments, our method outperforms the comparison methods in success rate, while also exhibiting lower collision checks and planning times. This suggests that our method is more efficient in avoiding unnecessary collisions and demonstrates superior reasoning capabilities. While maintaining high planning efficiency, our method is also capable of finding high-quality paths. Moreover, from the standard deviation, it is evident that our method exhibits more robust performance across different environments.
>
> Additionally, we further analyze the statistical significance of the differences between our method and the comparison methods for each metric (**:p<=0.01; *:p<=0.05; -:p>0.05). There are clear and significant differences between our method and the comparison methods across all four metrics in the vast majority of environments.
>
> #### Table 1: Comparison and significant differences in success rate under different environments
> |SR|stick3|link8|ur5|kuka13|kuka2Arms|kuka3Arms|
> |-|-|-|--|-|-|-|
> |RRT*|0.62±0.03(**)|0.41±0.06(**)|0.39±0.02(**)|0.67±0.09(**)|0.69±0.05(**)|0.25±0.07(**)|
> |BIT*|0.97±0.03(*)|0.93±0.03(-)|0.99±0.02(**)|1.00±0.00(**)|1.00±0.01(**)|0.57±0.00(**)|
> |LazySP|0.98±0.02(-)|0.88±0.03(**)|0.99±0.01(*)|1.00±0.02(**)|0.99±0.01(**)|0.56±0.00(**)|
> |NEXT|0.96±0.02(**)|0.41±0.04(**)|0.38±0.05(**)|0.61±0.02(**)|0.66±0.04(**)|0.37±0.13(**)|
> |GNN-Explorer|0.98±0.01(**)|0.92±0.03(**)|0.98±0.02(*)|1.00±0.02(-)|1.00±0.01(**)|0.57±0.00(**)|
> |GraphMP|0.97±0.05(**)|0.89±0.04(**)|0.96±0.02(**)|0.98±0.04(**)|0.98±0.02(**)|0.61±0.10(**)|
> |BrainyMP|0.97±0.03(*)|0.90±0.01(**)|0.99±0.02(**)|0.99±0.03(**)|0.99±0.01(**)|0.64±0.03(*)|
> |Our|0.98±0.01|0.93±0.02|0.99±0.02|1.00±0.01|0.99±0.02|0.64±0.02|
>
> #### Table 2 Comparison and significant differences in collision check under different environments
> |CK|stick3|link8|ur5|kuka13|kuka2Arms|kuka3Arms|
> |-|-|-|-|-|--|-|
> |RRT*|10150.45±61.71(**)|9351.92±71.21(**)|3142.53±6.79(**)|3001.12±21.74(**)|2818.19±38.72(**)|2996.24±25.08(**)|
> |BIT*|11357.34±341.89(**)|21419.08±937.17(**)|5418.59±490.37(**)|2060.64±974.84(**)|1787.27±176.63(**)|8127.88±117.93(**)|
> |LazySP|7707.78±127.51(**)|10410.65±106.11(**)|2775.13±97.89(**)|469.56±44.13(**)|567.75±28.30(**)|2699.10±92.57(**)|
> |NEXT|6241.26±115.26(**)|12979.51±45.84(**)|6406.80±99.50(**)|4908.82±33.22(**)|4577.05±51.34(**)|2226.26±70.56(**)|
> |GNN-Explorer|9022.12±268.86(**)|11242.59±335.63(**)|3014.54±193.82(**)|683.99±39.34(**)|605.37±36.39(**)|2913.49±31.52(**)|
> |GraphMP|7496.27±111.95(**)|13066.42±282.84(**)|2696.48±47.32(**)|627.05±11.14(**)|583.56±1.65(**)|1866.42±10.15(**)|
> |BrainyMP|6932.28±89.16(**)|9747.70±209.49(**)|1631.12±25.05(*)|401.06±5.04(*)|405.74±2.04(**)|1576.54±6.14(**)|
> |Our|5300.38±24.46|8029.68±178.82|1568.36±19.13|390.96±3.17|386.19±1.39|1256.47±12.24|
>
> #### Table 3:  Comparison and significant differences in path cost under different environments
> |PC|stick3|link8|ur5|kuka13|kuka2Arms|kuka3Arms|
> |-|-|-|-|-|-|-|
> |RRT*|1.37±0.01(**)|7.96±0.03(**)|4.11±0.03(**)|9.19±0.05(**)|9.69±0.03(**)|11.28±0.12(**)|
> |BIT*|1.73±0.02(**)|14.25±0.20(**)|11.45±0.15(**)|11.95±0.33(**)|12.05±0.06(**)|17.82±0.13(**)|
> |LazySP|1.84±0.01(**)|15.74±0.07(**)|11.95±0.16(**)|16.58±0.19(**)|16.12±0.11(**)|21.14±0.15(**)|
> |NEXT|1.23±0.01(**)|7.97±0.03(**)|4.85±0.11(**)|48.58±0.04(**)|10.10±0.09(**)|15.23±0.10(**)|
> |GNN-Explorer|2.02±0.01(**)|19.16±0.44(**)|12.70±0.20(**)|16.77±0.59(**)|16.71±0.14(**)|20.37±0.31(**)|
> |GraphMP|1.36±0.01(**)|10.62±0.04(**)|8.69±0.06(**)|11.15±0.17(**)|11.19±0.03(**)|16.70±0.34(**)|
> |BrainyMP|1.39±0.01(**)|9.77±0.16(**)|7.42±0.02(*)|9.31±0.07(-)|11.19±0.15(**)|16.64±0.31(-)|
> |Our|1.31±0.01|9.28±0.10|7.39±0.01|9.34±0.03|10.28±0.05|16.27±0.18|
>
> #### Table 4:  Comparison and significant differences in planning time under different environments
> |PT|stick3|link8|ur5|kuka13|kuka2Arms|kuka3Arms|
> |-|-|-|-|-|-|-|
> |RRT*|234.33±1.36(*)|1261.37±29.46(**)|295.52±2.44(**)|268.17±4.60(**)|195.42±6.99(**)|667.03±5.54(**)|
> |BIT*|355.06±10.87(**)|1418.83±43.52(**)|303.35±27.81(**)|211.94±38.90(**)|127.94±12.41(**)|1079.48±22.26(**)|
> |LazySP|894.47±36.23(**)|3160.85±42.49(**)|490.30±28.36(**)|118.93±23.53(**)|171.51±12.37(**)|855.14±10.64(**)|
> |NEXT|588.75±9.35(**)|19514.14±260.53(**)|5205.27±111.67(**)|3918.85±104.21(**)|3808.32±70.80(**)|4421.12±22.54(**)|
> |GNN-Explorer|692.75±26.34(**)|1936.34±39.33(**)|569.55±52.02(**)|122.85±11.97(**)|111.76±15.61(**)|769.18±6.06(**)|
> |GraphMP|501.94±13.28(**)|1621.73±40.14(**)|293.84±40.24(**)|133.99±6.65(**)|91.75±1.96(**)|410.42±2.87(**)|
> |BrainyMP|332.63±9.20(**)|961.48±43.42(**)|165.14±2.90(-)|76.49±1.30(**)|63.78±1.91(-)|403.44±8.42(**)|
> |Our|244.43±3.27|790.20±23.48|162.25±2.75|68.13±1.87|62.61±0.99|321.53±6.23|
>
> **Q1. Definition of random geometric graphs (RGGs)**
>
> Thank you for this valuable suggestion. We agree that formally defining the random geometric graphs (RGGs) upon their first mention is essential for reader clarity. We have added the following definition at line 36. "Specifically, GNN-based methods enhance search efficiency by operating on random geometric graphs (RGGs) constructed from sampled configurations, avoiding full workspace encoding [16]. Here, RGGs denote the undirected k-nearest neighbor (k-NN) graph built on free configuration space."
>
> **Q2. Iterations of obstacle encoding T**
>
> Thank you for pointing out he need for an ablation study on the number of iterations T in the Selector. Our choice of T=3 was based on precedent in related work [GNN-Explorer, GraphMP, BrainyMP], all of which apply three iterations for obstacle encoding on the same public dataset and with similar modules. Consequently, we followed prior work in setting the iteration count T and did not perform a dedicated ablation study. We have updated the Experimental Setting (Section 5.1) to include a detailed description of the iteration in the Perceptive Segment Selectors. We apologize that we are currently unable to train the Selector at different T values due to time constraints. In future work, we will conduct experiments varying T∈{1,2,3,5,10} to evaluate its impact on planning performance and to establish a stronger empirical basis for choosing this parameter.

---

> > ### Comment · Reviewer_snn6 · 2025-08-05
> >
> > First, I would like to sincerely thank the authors for their detailed and thoughtful responses to my concerns. I also wish to apologize for my delayed comment during this discussion period.
> >
> > The authors' rebuttal has addressed most of my initial concerns. However, the primary reason for my current score, "Marginal improvements in success rate" remains: I believe the validation and analysis of the method's performance are not yet sufficiently clarified.
> >
> > I understand the argument that existing methods already achieve near-perfect success rates on the chosen benchmarks, making it difficult to demonstrate a significant performance improvement. I also agree that the enhancement in efficiency is a valuable contribution. Nevertheless, a crucial point remains ambiguous: **Is the proposed method intended to  only yield improvements in efficiency, or can it also enhance performance, an aspect that could not be thoroughly analyzed due to the limitations of the benchmarks?**
> >
> > If the contribution is solely in efficiency, I believe a more in-depth analysis and discussion focusing on this aspect would be necessary. On the other hand (and I suspect this is the case), if the method is indeed capable of improving performance, I feel this should have been demonstrated on an alternative benchmark—even a simple one—where the performance ceiling is not already met. Additionally, for Kuka3Arms, there are quite huge rooms to be improved (the state-of-the-art success rate is about 64%).
> >
> > I fully recognize that incorporating such a revision is challenging within the limited discussion period. Clarifying this point, however, would share my remaining concern more clearly and make the paper's contributions significantly more concrete.
> >
> > Thank you for submitting your excellent research to our community.

---

> > > ### Author Response · Authors · 2025-08-06
> > >
> > > Thank you very much for your support and recognition of our revisions. We have provided point-by-point responses to your concerns below. We hope our replies will adequately address your concerns.
> > >
> > > **1. Contributions of NeuroMP**
> > >
> > > 1）Addressing BrainyMP's limitations by technical design
> > >
> > > NeuroMP and BrainyMP are built upon the GraphMP framework. Compared to GraphMP, they achieve notable improvements in planning efficiency (collision checks and planning time) and planning quality (path cost and success rate). However, BrainyMP has several limitations: (i) the Selector may inaccurately identify risky edges, leading to suboptimal graph construction; (ii) Under weakly connected graph structures, although subgraphs can partially enhance local–global relationships, their ability to capture global information is constrained by the number and coverage of subgraphs, limiting the overall structural reasoning capability. Moreover, extracting subgraphs incurs a high computational cost, and different subgraphs may contain redundant information.
> > >
> > > To address these limitations, NeuroMP introduces the following contributions: (i) a gated dual-branch Transformer architecture is adopted to improve the Selector’s collision probability prediction; (ii) spectral features and a dual-channel collaborative learning strategy (A2FL) are incorporated to enhance global topological awareness, thereby improving planning performance under weakly connected graph structures.
> > >
> > > 2) Improvements in performance and efficiency
> > >
> > > Compared to BrainyMP, a clear improvement in planning efficiency is achieved by NeuroMP, as evidenced by the reduced number of collision checks, which also indicates a lower sample requirement. Despite using fewer samples, NeuroMP maintains or even improves the path cost and success rate. Notably, in the Link8 environment, NeuroMP achieves a success rate of 0.93, compared to 0.90 for BrainyMP.
> > >
> > > In addition, the model parameters (Params) and FLOPs for NeuroMP and BrainyMP have been computed (detailed results and analysis can be found in Reviewer 7bRM’s response to Q2. Model Complexity). The model complexity of NeuroMP is significantly lower. For instance, in the Kuka2Arms environment, BrainyMP contains nearly 21% more parameters than NeuroMP.
> > >
> > > **2. Validation of other benchmarks**
> > >
> > > We have further validated the advantages of our method in real-world data CSM and the U-shaped environment (Link8).
> > >
> > > The results of GraphMP, BrainyMP, and NeuroMP on the CSM dataset are shown in Table 1. In the real-world setting, NeuroMP outperforms in planning efficiency and path quality.
> > > #### Table 1: Comparison on the CSM dataset
> > > |Method|SR|CK|PC|PCP|PT|
> > > |-|-|-|-|-|--|
> > > |GraphMP|0.86|1045.66|0.90|2.06|151.27|
> > > |BrainyMP|0.87|692.84|0.90|1.94|125.39|
> > > |**NeuroMP**|**0.88**|**526.42**|**0.88**|**1.86**|**102.78**|
> > >
> > > The results in the U-shaped environment are presented in Table 2. In the U-shaped environment (Link8), compared to GraphMP, NeuroMP demonstrates significant improvements in planning efficiency and path quality. When compared to SOTA, NeuroMP achieves the same success rate with fewer samples, while further optimizing path cost and planning efficiency.
> > > #### Table 2: Comparison in a U-shaped environment (Link8)
> > > |Methods|SR|CK|PC|PCP|PT|
> > > |--|--|--|--|--|-|
> > > |GraphMP|0.93|8085.31|8.23|9.94|556.76|
> > > |BrainyMP|**0.95**|6923.27|8.03|9.49|367.59|
> > > |**NeuroMP**|**0.95**|**5848.10**|**7.71**|**8.90**|**287.44**|
> > >
> > > **3. Reasons for poor performance in Kuka3Arms**
> > >
> > > The performance degradation in the Kuka3Arms environment may be attributed to: (i) the increased configuration space dimensions (14D → 21D), and (ii) the complexity of coordinating three robotic arms, which significantly increases planning difficulty, especially as each arm must avoid collisions with both other arms and static obstacles. Therefore, multi-arm manipulation scenarios are considered dynamic environments by some studies [1] due to the interactions between manipulators.
> > >
> > > Given the significant increase in difficulty in the Kuka3Arms environment, the success rates of NeuroMP and BrainyMP did not show substantial improvement, possibly due to the current "static" nature of the planning framework. In future work, we plan to extend our approach to better adapt to dynamic settings by: (i) implementing dynamic pruning strategies for RGGs, and (ii) incorporating time-based information and other dynamic adaptations.
> > >
> > > Thanks again for the valuable feedback and recognition of our work!
> > >
> > > ---
> > > [1] Zhang R., Y et. al. Learning-based motion planning in dynamic environments using gnns and temporal encoding. Advances in Neural Information Processing Systems, 2022, 35: 30003-30015.

---

> > > > ### Comment · Reviewer_snn6 · 2025-08-06
> > > >
> > > > I would like to thank the authors for their detailed comments regarding my concerns. **Their additional explanations have sufficiently addressed the issues I raised.**
> > > >
> > > > In particular, I was hoping for a clearer highlighting of the scenarios where NeuroMP holds a significant advantage in effectiveness. I now believe that their detailed explanation of planning efficiency, especially the lower sample requirement as exemplified by fewer collision checks, indirectly demonstrates this strength. I am convinced that if the number of collision checks were constrained, NeuroMP would likely exhibit a higher success rate compared to conventional methods.
> > > >
> > > > Furthermore, I find the verification on additional benchmarks and the analysis of performance limitations on the Kuka3Arms benchmark to be thorough and satisfactory.
> > > >
> > > > Therefore, I will be raising my score for this paper.

---

> > > > > ### Author Response · Authors · 2025-08-06
> > > > >
> > > > > We sincerely appreciate your valuable feedback and recognition of our revisions. These suggestions have significantly improved the quality of the manuscript. We will be sure to update the paper according to your suggestions.

---

### Official Review · Reviewer_7bRM · 2025-07-03

**Clarity:** 3
**Significance:** 3
**Originality:** 3
**Rating:** 6
**Confidence:** 3

**Summary:**

This paper introduces NeuroMP, a novel brain - inspired motion planning framework for high - dimensional continuous spaces. Modeling the two - stage Perception - Decision process of the human brain, NeuroMP incorporates a Perceptive Segment Selector and a Global Alignment Heuristic. The former constructs safer graphs, and the latter guides search in weakly connected graphs. Experimental results demonstrate NeuroMP's superiority in planning efficiency and path quality over existing methods while maintaining a high success rate.

**Questions:**

1. The experiments are conducted in static settings. How will you adapt NeuroMP to dynamic or real - world environments with uncertainties in the future? What modifications or extensions are needed for the model to handle these challenges?

2. Could you provide a more detailed analysis of NeuroMP's computational complexity and memory usage in different environmental settings? This would help assess its suitability for resource - constrained devices and guide further algorithmic optimization.

3. Have you considered combining NeuroMP with other advanced planning algorithms or architectures to leverage their respective advantages? For instance, integrating it with reinforcement - learning - based methods could enhance its decision - making and adaptability in complex environments.

**Ethical Concerns:**

["NO or VERY MINOR ethics concerns only"]

**Final Justification:**

Overall, the authors have addressed some of my concerns, so I will increase my score.

**Limitations:**

Yes

**Paper Formatting Concerns:**

No formatting issues.

**Quality:**

3

**Strengths And Weaknesses:**

### Strengths
1. NeuroMP, simulating the brain's two - stage model, offers a new solution for high - dimensional motion planning.
2. NeuroMP outperforms other advanced planners in success rate, collision checks, and planning time across various scenes, proving its better planning efficiency and path quality.
3. The Perceptive Segment Selector accurately identifies environment patterns to build safer graphs. The Global Alignment Heuristic, with spectral features and A2FL, enhances structural reasoning and heuristic estimation.

### Weaknesses
1. As NeuroMP searches for paths on sampled random geometric graphs, it cannot ensure the existence of the optimal path within the constructed graph, limiting its application in optimal - path - critical scenarios.
2. Experiments mainly focus on specific static environments like mazes and robotic - arm operation tasks. The model's generalization ability in dynamic or real - world settings remains unverified.

---

> ### Author Rebuttal · Authors · 2025-07-30
>
> Thank you very much for your positive comment regarding our work as "novel brain-inspired framework". We also appreciate your valuable suggestions. Following your suggestions, we provide point‑by‑point responses below and hope they adequately address your concerns.
>
> ### For Weaknesses:
>
> **W1. Limitation of the optimal path**
>
> We fully agree that searching on a sampled random geometric graph (RGG) cannot guarantee inclusion of the true optimal path, thereby limiting theoretical optimality. Although our selective sampling strategy mitigates this issue to some degree, it cannot ensure optimal‑path coverage. To address this, we will (i) design enhanced graph‑construction strategies that leverage prior knowledge to guide sampling, reduce randomness, and increase the probability that optimal‑path structures are represented; (ii) Exploring environment‑aware dynamic pruning, which can adaptively refine the RGG online to improve coverage of critical connections and ensure robust path representation.
>
> **W2. Generalization are not fully evaluated**
>
> We appreciate your comment on limitations of our experimental settings and the need to verify generalization in dynamic or real‑world scenarios. Our current evaluation focuses on static environments. Following your suggestion, we have added a focused discussion of the model’s generalization to dynamic and real‑world settings and clarified how these considerations inform our future extensions and evaluations.
>
> 1. Generalization of real-world environments
>
> Real-world scenarios or simulations that mimic real-world conditions can further validate the effectiveness of our method. Following your suggestion, we select the City/Street Map (CSM) Dataset as the real-world scenario for validation. We have included a description of the dataset, experimental results, and analysis.
>
> CSM. Sturtevant et al.[1] used drones to collect 30 real city maps with marked obstacles represented as binary images. Based on this, we randomly generate 2000 training maps and 1000 test instances from the 30 maps. For each map, the start and goal are randomly generated in the free space. These maps present significantly higher complexity than synthetic environments: irregular obstacle shapes, narrow alleyways, and dense urban structures that challenge traditional planners.
> #### Table 1 Comparison of all methods on the CSM dataset
> |Method|SR|CK|PC|PCP|PT|
> |-|-|-|-|-|--|
> |RRT*|0.82|1609.79|0.97|2.61|112.99|
> |BIT*|0.86|1262.28|0.94|2.24|363.39|
> |LasySP|0.85|662.99|0.98|2.31|206.71|
> |NEXT|0.79|6552.28|0.90|2.78|2590.84|
> |GNN-Explorer|0.85|805.70|1.04|2.38|362.55|
> |GraphMP|0.86|1045.66|0.90|2.06|151.27|
> |BrainyMP|0.87|692.84|0.90|1.94|125.39|
> |**NeuroMP**|**0.88**|**526.42**|0.88|1.86|**102.78**|
> |GNN-Explorer-S|0.85|951.39|0.95|2.00|373.34|
> |GraphMP-S|0.86|1064.91|0.89|1.91|153.96|
> |BrainyMP-S|0.87|714.79|0.89|1.80|127.77|
> |**NeuroMP-S**|**0.88**|545.07|**0.87**|**1.74**|105.02|
>
> The results on the CSM dataset are demonstrated in Table 1. In real-world environments with fine-grained and densely distributed obstacles, robots typically require more collision checks and longer planning times. NeuroMP achieves the highest success rate (0.88) while requiring 67% fewer collision checks than RRT* and 87% faster planning than BIT*. Due to encoding the entire workspace, NEXT incurs substantially higher computational overhead compared to other methods. In contrast, our method not only achieves competitive success rates but also the fewest collision checks and the shortest planning time across all baselines. These results demonstrate superior real-world generalization of GNN-based approaches, particularly ours, offering excellent stability in complex urban environments.
>
> [1] N. R. Sturtevant, “Benchmarks for grid-based pathfinding,” IEEE Trans. Comput. Intell. AI Games, vol. 4, no. 2,pp. 144–148, 2012
>
> 2. Generalization of dynamic environments
>
> Our study uses strict static environments to validate the method. However, multi-arm manipulation scenarios (e.g., Kuka2Arms, Kuka3Arms) are considered dynamic environments by the work [2] due to the interactions between manipulators. Coordinating multiple manipulators to avoid collisions with each other and with static obstacles poses significant challenges. This both highlights our framework’s partial applicability to dynamic settings and underscores the need to handle moving obstacles. In future work, we will (i) design dynamic pruning strategies for RGGs to adapt to real-time changes in the environment, and (ii) encode temporal information into the graph and heuristic to model time‑varying constraints and interactions.
>
> We also recognize the need for broader validation on more complex real‑world datasets. We will therefore expand the experimental scope to comprehensively assess generalization.
>
> [2] Zhang R., Y et. al. Learning-based motion planning in dynamic environments using gnns and temporal encoding. Advances in Neural Information Processing Systems, 2022, 35, 30003-30015.
>
> ### For Questions:
> **Q1. How will NeuroMP adapt to dynamic or real‑world environments**
>
> We appreciate the insightful question. Adapting NeuroMP to uncertainty in dynamic and real‑world settings will be our future focus, with two main extensions:
>
> 1. Dynamic environments: (i) Architecture. Integrate a temporal perception module that learns and predicts scene changes, updating the random geometric graph (RGG) online. (ii) Adaptive Replanning. Trigger replanning when significant dynamics occur, using RGG pruning and path‑rewiring guided by historical decisions to maintain feasibility in real time.
>
> 2. Real‑world environments: (i) Dataset Expansion. Assemble a diverse, disturbance‑rich dynamic dataset (e.g., moving obstacles, sensor noise) with augmentation to improve generalization. (ii) RL Integration. Employ reinforcement learning to refine planning strategies via reward‑driven interaction, enabling online adjustment based on observed performance.
>
> These enhancements will be detailed in the revised manuscript’s Future Work section.
>
> **Q2. Model Complexity**
>
> We sincerely appreciate your insightful question. Computational complexity and memory consumption are indeed critical factors in evaluating the applicability of a model on resource-constrained devices. Your suggestion provides valuable guidance for improving our analysis.
>
> 1. Parameters and FLOPs
>
> We computed the number of model parameters (Params) and floating-point operations (FLOPs) for NeuroMP and learning-based baselines across different environments in Table 2. For NEXT, FLOPs increase dramatically with workspace dimensionality, and Params grow modestly with configuration-space dimension. For GNN-based planners, Params and FLOPs scale with the size of the processed RGG and the configuration-space dimension. To further assess scalability, we will compare these metrics across varying RGG sizes in future work. Overall, NeuroMP is more efficient than NEXT and GNN-Explorer, and competitive with GraphMP/BrainyMP, while delivering superior performance. This demonstrates that our model maintains a good balance between model capacity and computational cost, making it well-suited for deployment in resource-constrained scenarios.
> #### Table 2 Parameters comparison
> |Method|Stick3|link8|ur5|kuka13|kuka2arms|kuka3arms|
> |-|-|-|-|--|-|-|
> |NEXT|0.1644M|0.1645M|0.3959M|0.3968M|0.3969M|0.3978M|
> |GNN-E|0.0911M|0.3580M|0.0925M|0.0952M|0.0956M|0.0982M|
> |GraphMP|0.0694M|0.0711M|0.0711M|0.0731M|0.0734M|0.0759M|
> |BrainyMP|0.0868M|0.0885M|0.0881M|0.0906M|0.0909M|0.0934M|
> |Our|0.0707M|0.0726M|0.0721M|0.0748M|0.0753M|0.0779M|
>
> #### Table 3 FLOPs comparison
> |Method|Stick3|link8|ur5|kuka13|kuka2arms|kuka3arms|
> |-|-|-|-|-|-|-|
> |NEXT|0.3775G|0.3375G|10.5719G|10.5719G|10.5719G|10.5719G|
> |GNN-E|0.0626G|0.2493G|0.0631G|0.0641G|0.0643G|0.0654G|
> |GraphMP|0.0538G|0.0546G|0.0543G|0.0555G|0.0557G|0.0568G|
> |BrainyMP|0.0622G|0.0538G|0.0664G|0.0704G|0.0706G|0.0718G|
> |Our|0.0609G|0.0622G|0.0615G|0.0627G|0.0629G|0.0642G|
>
> 2. Optimization for resource-constrained devices
>
> Although our framework comprises multiple stages, each module is individually lightweight and simple. Nonetheless, several optimization directions can be considered for deployment on resource-limited platforms: (i) incorporating sparse graph construction strategies to reduce overall complexity; (ii) employing quantized search algorithms (e.g., integer-based A* variants) to reduce floating-point computation; (iii) applying low-rank decomposition to compress node feature representations and reduce memory footprint; and (iv) dynamically releasing unused historical memory during local updates to improve memory efficiency.
>
> **Q3. Future integration of NeuroMP with other methods**
>
> We appreciate your suggestion and agree that integrating NeuroMP with other planning paradigms is a fruitful avenue. In particular, coupling with reinforcement learning (RL) could address NeuroMP’s limitations under dynamic uncertainty and multi‑objective trade‑offs. Specifically, we plan to explore: (i) Policy Optimization. RL can be used to train robots that adaptively adjust graph sampling density or node feature weighting based on environmental characteristics (e.g., obstacle distribution or dynamic frequency), enabling the constructed graph to better match real-time planning demands and improve efficiency and quality. (ii) Hybrid Decision Framework. Deploy NeuroMP for detailed local search while an RL agent oversees global strategy, selecting pivotal nodes, resolving competing objectives, or redirecting the search in rapidly evolving scenes. (iii) Experience‑Driven Adaptation. Store NeuroMP-generated paths as experience buffers and train an RL module to recall and reuse them when facing similar scenarios, reducing redundant exploration and speeding up replanning.

---

> > ### Comment · Reviewer_7bRM · 2025-08-05
> >
> > I thank the authors for their thoughtful and detailed responses. The clarifications have addressed my questions and helped me better understand the motivations and design choices in the paper. I appreciate the efforts to improve the manuscript based on the feedback, and I maintain my positive assessment of the work.

---

> > > ### Comment · Area_Chair_vbDu · 2025-08-06
> > >
> > > Dear reviewer, thank you for engaging in the discussion

---

> ### Author Response · Authors · 2025-08-05
>
> We sincerely appreciate the constructive comments and positive assessment. We will update the manuscript following your suggestions.

---

### Official Review · Reviewer_d3Rn · 2025-07-03

**Clarity:** 1
**Significance:** 2
**Originality:** 3
**Rating:** 4
**Confidence:** 5

**Summary:**

This paper presents NeuroMP, a two-stage brain-inspired motion planning framework designed to enhance planning performance in high-dimensional spaces. In the first stage, the method constructs a graph through selective sampling, removing edges that violate constraints through edge selection to create a safe graph. Finally, node heuristics are predicted for this safe graph to determine the path solution that connects the start and goal. Across various motion planning tasks, NeuroMP demonstrates superior planning efficiency and path quality, highlighting the potential of brain-inspired methods to improve robotic decision-making.

**Questions:**

How do these methods compare to algorithms like P-NTFields, which do not require expert demonstrations but exhibit high inference efficiency and success rates?

Is the brain-inspired approach genuinely beneficial in this context? The computational times are relatively higher than those observed in many end-to-end learning-based methods like MPNet and NTFields. Is the brain-inspired approach helping or is it acting as a limiting factor? This needs to be evaluated.

Does the method recover from failure? What would be the pathway to address failure cases? Where did the method generally fail?

Learning-based methods often struggle with convoluted U-shaped environments. For instance, consider a robot manipulator moving from one shelf to another, where the start and goal positions are deep inside those shelves. Have you tested your approach in those scenarios?

The overall performance gain is very marginal over brainyMP. What were the key components that contributed to the relatively improved performance?

Why did performance significantly drop in the Kuka 3 tests?

The performance variations between all ablative approaches are marginal. For instance, is shortcut retrieval really necessary?

**Ethical Concerns:**

["NO or VERY MINOR ethics concerns only"]

**Final Justification:**

Overall, the authors have addressed some of my concerns, so I will increase my score. However, I believe the paper requires major revisions, particularly in the areas of writing and the literature review.

**Limitations:**

Yes

**Paper Formatting Concerns:**

The repeated analogy between the brain and the proposed algorithm is unnecessary and negatively impacts the flow for the reader. Although the method is straightforward, it is presented in a convoluted manner. The paper dedicates excessive space to discussing the relationship between brain function and the proposed approach. A simple, intuitive description would have sufficed, allowing for a more thorough literature review and better presentation of the results.

**Quality:**

2

**Strengths And Weaknesses:**

**Strengths:**

The paper is technically sound. The approach is multistaged, making progress step by step towards finding solutions. Experiments demonstrate improvements in some metrics compared to prior methods.

**Weaknesses:**
The idea presented in this paper shares similarities with many existing learning-based approaches for motion planning; however, these works are neither cited nor compared. These methods also plan in stages: For instance, "Motion Planning Transformers" and "Learning Sampling Dictionaries for Efficient and Generalizable Robot Motion Planning with Transformers." The literature review is generally lacking and misses several key contributions in learning-based motion planning, such as MPNet, NeuralMP, NTFields, P-NTFields, and MPiNet.

The repeated analogy between the brain and the proposed algorithm is unnecessary and negatively impacts the flow for the reader. Although the method is straightforward, it is presented in a convoluted manner. The paper dedicates excessive space to discussing the relationship between brain function and the proposed approach. A simple, intuitive description would have sufficed, allowing for a more thorough literature review and better presentation of the results.

Additionally, there is no demonstration or illustration of task complexity, making it difficult to gauge the efficacy of the proposed method. The paper would benefit from including real-world experiments.

Furthermore, there is no discussion on the training times or the time required to collect the dataset.

---

> ### Author Rebuttal · Authors · 2025-07-30
>
> Thank you very much for your valuable feedback and for pointing out the weaknesses in our work. We apologize for the unclear statement that may have caused some misunderstanding. To clarify, the planning time refers to the total time taken to complete 1000 tasks, rather than the average time for a planning problem. Following your suggestions, we have provided detailed point-by-point responses below.
>
> **W1. Literature review and comparisons**
>
> We apologize for the incomplete overview of learning-based motion planning methods due to space limitations. As a result, we focused primarily on the most relevant work related to high-dimensional continuous motion planning. We agree that a more detailed discussion of the related work would indeed provide readers with a better understanding of the field's progress. Following your suggestion, we plan to incorporate the seven works you mentioned into the related work section under learning-based sampling methods. Additionally, we have already included direct comparison results and analysis with P-NTFields (Q1).
>
> **W2. Simplifying presentation & brain analogy**
>
> We appreciate your valuable suggestions on improving the readability and structure of the paper. In response to your concerns:
>
> 1. Simplification of the brain analogy. In the revised manuscript, we have implemented the following changes: (i) We briefly mention the perception-decision inspiration in the introduction and removed neural mechanism analogies that are less relevant to the core method to avoid unnecessary complexity. (ii) In the method section, we focus on the design motivation and modules, without reiterating the brain analogy.
>
> 2. Expansion of literature review and experimental analysis. We have included additional results and analyses, such as comparisons in **real-world environments** (detailed results and analysis can be found in the response to Reviewer 7bRM under "W2"), the **U-shape environment** (Q4), and with **comparison methods** (Q1).
>
> **W3. Task complexity description**
>
> To clarify the performance of our method across tasks of varying difficulty, we quantify the complexity of the datasets in Table 1. Except for Kuka3Arms, all datasets are publicly available, with Stick3 derived from NEXT and the others sourced from GNN-Explorer. By using publicly available datasets, we avoid the time-consuming process of data collection and annotation, ensuring a more efficient and fair comparison with similar methods. Detailed descriptions of each dataset have been added to the Appendix.
> #### Table 1 Datasets complexity
> |Environment|Obstacles|Map Size|Robot|Configuration Dimension|
> |-|-|-|-|-|
> |Stick3|>46%|15*15|A 3DoF robot|3D|
> |Link8|>46%|15*15|A 8DoF robot |8D|
> |Ur5|Two sets of boxes, poles, pads|15 * 15 * 15|A 6DoF UR5 robot|6D|
> |Kuka13|Some boxes|15 * 15 * 15|A13DoFKUKAarm|13D|
> |Kuka2Arms|Some boxes|15 * 15 * 15|Two 7DoF KUKA arms|14D|
> |Kuka3Arms|Some boxes|15 * 15 * 15|Three 7DoF KUKA arms|21D|
>
> **W4. Discussion on the training times and model complexity**
>
> Thank you for your valuable feedback. Training time is an important factor when evaluating the practicality and reproducibility of methods. We computed the average time required to train two models (the Selector and Heuristic) for one epoch in Table 2, the training time is mainly proportional to the complexity of the environment. The time cost for training our models is relatively modest.
> #### Table 2 Training times (mm:ss)
> ||Stick3|link8|ur5|kuka13|kuka2arms|kuka3arms|
> |-|-|-|-|-|-|-|
> |Selector|00:44|01:32|03:30|02:37|03:46|04:02|
> |Heuristic|02:11|03:55|04:03|05:20|04:34|07:11|
>
> Model complexity. Detailed experimental results and analysis can be found in the response to Reviewer 7bRM "Q2".
>
> **Q1. Comparison with P-NTFields**
>
> Thank you for providing the new method. The advantage of P-NTFields is that it avoids the computation-heavy training process by not requiring expert demonstrations. In the P-NTFields's Ur5 environment, a cabinet with narrow passages was chosen. Its path planning mainly involves obstacle avoidance in a 2D plane. In contrast, our UR5 environment involves a 6-DOF robotic arm that not only needs to handle obstacles in 3D space but also considers the fixed base and movement limitations of the robot's joints. The configuration space includes the 3D workspace and the DoF of the robot arm. Therefore, our UR5 environment is more complex. This difference is further validated by the results of RRT*:RRT* achieves a 0.39 success rate in our UR5 environment, compared to a 0.67 success rate in the P-NTFields's UR5 environment. The results for P-NTFields in our UR5 environment are shown in Table 3, where PT refers to the total time taken to complete 1000 tasks. The success rate and inference efficiency of P-NTFields show a noticeable decline. Additionally, the number of parameters in our model is approximately 15% of that required by P-NTFields. Although our method is multi-stage, it is lightweight, simple, and effective.
> #### Table 3 Comparison with P-NTFields
> |Methods|Params|SR|CK|PC|PCP|PT|
> |-|-|-|-|-|-|-|
> |Our|0.0721M|0.99|1547.76|7.37|7.70|158.24|
> |P-NTField|0.5761M|0.54|1717.42|12.92|19.09|768.04|
>
> **Q2&Q5&Q7. Discussion on component performance improvement**
>
> 1. Contributions of components and ablation differences
>
> The contributions of the components are as follows: (i) Selective Sampling initially optimizes graph construction, bringing it closer to the optimal solution space. (ii) The Selector effectively removes collision-risk edges from the original graph, providing a safe graph for subsequent searches and simplifying computations. (iii) The Heuristic provides reliable heuristic values to A*, optimizing search efficiency and path quality. (iv) Shortcut Retrieval can refine the searched path, resulting in further optimization of path cost in a minimal amount of time.
>
> The performance improvement of the method does not rely on a single breakthrough from one component but rather on the synergistic optimization of different components, which enables better robustness and efficiency in complex high-dimensional scenarios. This gain might manifest as small differences in simpler scenarios but becomes significantly amplified in extreme cases. For instance, in extremely weak graph structures, our method shows a marked improvement in success rates compared to GraphMP, further optimizing planning efficiency and path quality.
>
> 2. Performance differences with BrainyMP
>
> In saturated benchmark tests, further improvements in success rate and path cost are limited, with the main contribution of NeuroMP being in efficiency. Enhancing planning efficiency is crucial for practical applications, especially in real-time domains, where reduced planning time and collision checks improve system responsiveness and stability. We compared BrainyMP and NeuroMP in weak graph structures. Tables 4 and 5 show the retained edge ratio and planning performance for both methods. In the Stick3 environment, both selectors perform similarly. However, NeuroMP generates more accurate heuristics, significantly improving planning efficiency. In the more complex Kuka13 environment, NeuroMP’s selector better identifies dangerous edges, slightly increasing planning efficiency while finding higher-quality paths.
> #### Table 4  BrainyMP and NeuroMP in Stick3
> |Method|Ratio|SR|CK|PC|PCP|PT|
> |-|-|-|-|-|-|-|
> |NeuroMP|0.63|0.97|6664.90|1.4|1.74|245.35|
> |BrainyMP|0.69|0.96|7300.75|1.41|2.12|270.86|
> #### Table 5 BrainyMP and NeuroMP in Kuka13
> |Method|Ratio|SR|CK|PC|PCP|PT|
> |-|-|-|-|-|-|-|
> |NeuroMP|0.46|0.91|624.66|10.02|12.75|117.50|
> |BrainyMP|0.26|0.52|543.24|10.29|24.29|106.85|
>
> **Q3. How to address failure cases**
>
> Our method checks for collisions during the planning process by evaluating point and edge collisions. If a collision occurs, points are resampled or alternative edges are selected. If the goal region is not reached within the specified number of samples, the planning is considered a failure, and no post-processing is applied to failed paths. Future work will focus on: (i) Retrieving and trimming collision path segments, such as resampling near obstacles; (ii) Designing improved graph construction strategies that incorporate prior knowledge to reduce risky edge generation; (iii) Exploring dynamic pruning strategies for RGGs based on environmental features. Failure Causes: (i) Maze: Large map sizes, high obstacle densities, and high-DoF in robots may cause path failures. (ii) Robotic Arm: the number of robotic arms, their DoF, and obstacle placements affect path planning success.
>
> **Q4. Discussion of the U-shaped environment**
>
> The U-shaped environment, a common challenge in motion planning, has been added to the revised manuscript. In a 15×15 2D workspace, U-shaped obstacles simulate narrow passages between shelves, with the start point near the bottom inside the U-shaped obstacle and the goal outside. Tests are conducted with stick robots across 500 U-shaped map instances. Our method shows significant advantages in reasoning efficiency, success rate, and path quality. Due to time constraints, we are unable to test more complex robots, but we will explore the U-shaped environment further in future work.
>
> |Method|SR|CK|PC|PCP|PT|
> |-|-|-|-|-|-|
> |RRT*|0.60|7290.00|1.44|6.81|84.34|
> |BIT*|1.00|4704.71|1.62|1.62|60.04|
> |LazySP|0.99|1856.48|1.73|1.78|31.61|
> |NEXT|0.99|2051.83|1.42|1.68|115.14|
> |GNN-Explorer|1.00|1419.12|2.19|1.78|33.77|
> |Graphmp|0.99|1791.17|1.68|1.77|40.08|
> |BrainyMP|0.99|1276.75|1.48|1.60|27.03|
> |NeuroMP|0.99|**1132.01**|**1.46**|**1.55**|**24.72**|
>
> **Q6. Performance decline in the Kuka3Arms**
>
> The performance degradation in the Kuka3Arms environment may be due to: (i) Increased Configuration Space Dimensions(14D->21D). (ii)Coordinating three robotic arms increases the planning difficulty, especially since each arm needs to avoid collisions with both other arms and obstacles. (iii) The new arm makes collision avoidance more challenging.

---

### Official Review · Reviewer_3DsE · 2025-07-04

**Clarity:** 3
**Significance:** 2
**Originality:** 3
**Rating:** 5
**Confidence:** 3

**Summary:**

This paper proposes to revisit ML-based path planning by designing a new pipeline inspired from human cognitive mechanisms, implementing a two-stage Perception-Decision model. The perception stage constructs an RGG using Transformer blocks while the decision stage uses a dual spatial and spectral RGG encoding to inform an A*-variant search. Results are presented on a number of robotics path planning benchmarks, reporting progress over the SotA.

**Questions:**

Questions:
Why are spectral features used at the decision rather than the perception part of the pipeline?
In the ablation study, the contribution of spectral features appears to vary significantly with the benchmarks considered: could you elaborate on this?
Why have you not considered standard path planning benchmarks [Sturtevant, 2012]?

Sturtevant, N.R., 2012. Benchmarks for grid-based pathfinding. IEEE Transactions on Computational Intelligence and AI in Games, 4(2), pp.144-148.

**Ethical Concerns:**

["NO or VERY MINOR ethics concerns only"]

**Final Justification:**

The authors provided an extremely detailed rebuttal which answered my most important points. The results on the named additional benchmarks were particularly convincing at short notice. Answers on spectral aspects were equally satisfactory.
The concern over static environments remains, however the proposed paper reframing under "limited resources" should improve its significance and perception.
Having considered other reviews but most importantly the similar quality of responses to other reviewers I have decided to upgrade my rating for the paper.

**Limitations:**

Limitations:
Although not specifically designated as a limitations section the discussion of subsections 5.4-5.6 addresses transparently and appropriately a number of limitations.

However, one major limitation of the paper is the imbalance between the effort and complexity of the method and the restriction to static environments. It might also defeat the brain metaphor, as cognitive systems would be primarily intended for dynamic environments and aim at horizon-bound, suboptimal, dynamic path planning (more on the RTA* philosophy).

**Quality:**

3

**Strengths And Weaknesses:**

Strengths:
The paper reports compelling results from comprehensive tests, comparing it to other SotA path planners (both ML planners and classical planner), on a number of benchmarks.
The text is very rigorously organized and presented, which partly compensates for the complexity of the contents. Highlights include balance between various sections and the level of detail of results presentation and ablation studies.

Weaknesses:
In my view the main weakness of this paper rests with the overall complexity of the approach, which may preclude its actual deployment on two levels: replicability and scalability. In that sense it would constitute more a ‘significance’ weakness than a ‘technical soundness’ weakness. Perhaps the overall architecture should be better justified rather than incrementally constructed (and somehow discussed later through the ablation study).
Other aspects, like restriction to static environments may constitute more a limitation than an intrinsic weakness at this stage of the research development.

---

> ### Author Rebuttal · Authors · 2025-07-30
>
> Thank you very much for your positive comment regarding our work as "compelling results" and "rigorously organized and presented". We also greatly appreciate your insightful comments pointing out the weaknesses in our work. Following your suggestions, we have provided detailed point-by-point responses below. We hope our replies will adequately address your concerns.
>
> ## For weaknesses:
> **W1. Replicability and scalability**
>
> We sincerely thank you for pointing out this concern. Our method adopts a multi-stage framework, which is indeed more complex compared to single-stage approaches. We discuss replicability and scalability from four aspects: technical reproducibility, model complexity, scalability, and real-world deployment.
>
> 1. Technical reproducibility
>
> To ensure reproducibility, we have released the full source code, environmental configurations, and datasets. Additionally, Section 5.1 of the paper provides detailed descriptions of parameter settings, which collectively guarantee the feasibility of reproducing our results.
>
> 2. Model complexity
>
> Although our framework consists of multiple stages, each module is individually simple and lightweight. We evaluated parameters (Params) and floating-point operations (FLOPs) for NeuroMP and learning-based baselines. Detailed experimental results and analysis can be found in the response to Reviewer 7bRM "Q2". NeuroMP is more efficient than NEXT and GNN-Explorer, and competitive with GraphMP and BrainyMP, while delivering superior performance. This balance is suitable for resource-constrained deployment.
>
> 3. Scalability & Real-world generalization
>
> Real-world scenarios or simulations that mimic real-world conditions can further validate the effectiveness of our method. Following your suggestion, we select the City/Street Map (CSM) Dataset as the real-world scenario for validation.
>
> CSM dataset. Sturtevant et al. used drones to collect 30 real city maps with marked obstacles represented as binary images. Based on this, we randomly generate 2000 training maps and 1000 test instances from the 30 maps. For each map, the start and goal are randomly generated in the free space. These maps present significantly higher complexity than synthetic environments: irregular obstacle shapes, narrow alleyways, and dense urban structures that challenge traditional planners.
> #### Table 1 Comparison of all methods on the CSM dataset
> |Method|SR|CK|PC|PCP|PT|
> |-|-|-|-|-|--|
> |RRT*|0.82|1609.79|0.97|2.61|112.99|
> |BIT*|0.86|1262.28|0.94|2.24|363.39|
> |LasySP|0.85|662.99|0.98|2.31|206.71|
> |NEXT|0.79|6552.28|0.90|2.78|2590.84|
> |GNN-Explorer|0.85|805.70|1.04|2.38|362.55|
> |GraphMP|0.86|1045.66|0.90|2.06|151.27|
> |BrainyMP|0.87|692.84|0.90|1.94|125.39|
> |**NeuroMP**|**0.88**|**526.42**|0.88|1.86|**102.78**|
> |GNN-Explorer-S|0.85|951.39|0.95|2.00|373.34|
> |GraphMP-S|0.86|1064.91|0.89|1.91|153.96|
> |BrainyMP-S|0.87|714.79|0.89|1.80|127.77|
> |**NeuroMP-S**|**0.88**|545.07|**0.87**|**1.74**|105.02|
>
> The results on the CSM dataset are demonstrated in Table 1. In real-world environments with fine-grained and densely distributed obstacles, robots typically require more collision checks and longer planning times. NeuroMP achieves the highest success rate (0.88) while requiring 67% fewer collision checks than RRT* and 87% faster planning than BIT*. Due to encoding the entire workspace, NEXT incurs substantially higher computational overhead compared to other methods. In contrast, our method not only achieves competitive success rates but also the fewest collision checks and the shortest planning time across all baselines. These results demonstrate superior real-world generalization of GNN-based approaches, particularly ours, offering excellent stability in complex urban environments.
>
> Our method requires further validation in real-world deployment scenarios, such as robotic arm manipulation in physical environments. For future work, we plan to evaluate our approach under more realistic settings, including robotic arm manipulation, mobile robot operations, and spatial navigation tasks.
>
> **W2. Restriction to static environments**
>
> We agree that the static environment focus constitutes a conscious limitation at this research stage rather than an inherent framework flaw. This scope is justified by its critical value in applications like robot-assisted surgery and spatial navigation, where our method establishes essential baselines for high-dimensional manipulation tasks. The demonstrated performance gains validate its effectiveness within this domain. In the revised Introduction, we have strengthened this context as you noted. Our method establishes reproducible benchmark conditions for real-world navigation and high-dimensional robotic manipulation tasks, and the observed performance improvements demonstrate its effectiveness.
>
> ## For questions:
> **Q1. Utilization stage of spectral features**
>
> 1. Alignment between spectral features and the requirements of the decision stage
>
> The perception stage aims to capture local features, such as node position and local neighborhood structures. Introducing spectral features at this stage may introduce global information prematurely, potentially blurring local details and impairing the accurate recognition of specific environmental elements. In contrast, the decision-making stage requires reasoning from a global perspective, such as evaluating the overall cost of different paths or assessing the long-term effectiveness of strategies. Spectral features are well-suited to support decisions by compensating for the limitations of local features in capturing long-range dependencies and topological relationships.
>
> 2. Computational efficiency and progressive information processing
>
> Inspired by the brain's two-stage Perception–Decision model, our framework adopts a progressive design that transitions from local perception to global decision-making: (i)During the perception stage, only local information is processed, resulting in low computational complexity and enabling a rapid initial understanding of the environment; (ii) In the decision stage, since redundant information has already been filtered during the perception stage, the computational burden associated with spectral feature extraction is significantly reduced. If spectral features are introduced directly during the perception stage, their high spatial dimensionality would result in substantial computational overhead, thereby compromising the practical efficiency of the method.
>
> **Q2. The contribution of spectral features**
>
> The significance of spectral features is in capturing global structural patterns, which is boost performance most in complex environments. In relatively simple environments (such as Kuka13 and Kuka3Arms), where obstacle density is low and planning goals are clear, local features (spatial features) are generally sufficient for effective decision-making. In such cases, the complementary effect of spectral features is limited, resulting in lower impact. However, in more complex environments like Link8, which exhibit high obstacle density and increased degrees of freedom, local features are prone to falling into "local optima." Spectral features significantly enhance global reasoning by encoding global topological properties through spectral distributions, thereby offering greater contributions under such challenging conditions.
>
> **Q3. Dataset selection**
>
> Our work focuses on high-dimensional continuous environments. Therefore, we adopt a publicly available dataset specifically designed for such settings, which has been widely adopted benchmarks used by NEXT, GNN-Explorer, GraphMP, and BrainyMP. This ensures a fair and consistent benchmark for performance comparison. On the other hand, the environments in [Sturtevant, 2012] are predominantly 2D discrete spaces, which differ significantly in structure and complexity from the problem settings we aim to address. As such, they may not be well-suited for evaluating motion planning methods in high-dimensional continuous spaces.
>
> ## For limitations:
>
> We sincerely thank you for pointing out this limitation. As discussed in Appendix G, we acknowledge the concern regarding the potential mismatch between the method’s complexity and its current application in static environments. We address this issue from two perspectives:
>
> 1. Imbalance between method complexity and static environment constraints
>
> Our study uses strict static environments to validate the method. However, multi-arm manipulation scenarios (e.g., Kuka2Arms, Kuka3Arms) are considered dynamic environments by some works [1] due to the interactions between manipulators. Coordinating manipulators to avoid collisions with each other and static obstacles partially demonstrates the framework’s applicability to dynamic environments. In future work, we plan to extend our approach to dynamic settings (including moving obstacles) by (i) implementing dynamic pruning strategies for RGGs and (ii) using limited-horizon planning techniques like RTA* to improve decision-making efficiency.
>
> [1] Zhang R., Y et. al. Learning-based motion planning in dynamic environments using gnns and temporal encoding. Advances in Neural Information Processing Systems, 2022, 35, 30003-30015.
>
> 2. Justification of the brain-inspired analogy
>
> We agree with your observation on suboptimal, horizon-limited planning under dynamic conditions. The brain's two-stage Perception–Decision model excels in making efficient suboptimal decisions under incomplete information, which aligns with methods like RTA* that prioritize real-time responsiveness. We have revised the paper to focus on decision-making under limited computational resources rather than framing the method as a structural emulation of brain regions in static settings.

---

> > ### Comment · Reviewer_3DsE · 2025-08-05
> > **Acknowledgement of rebuttal**
> >
> > I thank the authors for their comprehensive response to my review and questions.
> > I found the response entirely satisfactory and on the only point not fully addressed, the response to the limitations section and in particular point 2. do allay my concerns. I have thus raised my score out of consistency.

---

> > > ### Author Response · Authors · 2025-08-05
> > >
> > > Thanks again for the valuable feedback and recognition of our work! We will be sure to update the manuscript according to your suggestions.

---

> > > > ### Comment · Area_Chair_vbDu · 2025-08-06
> > > >
> > > > Dear reviewer,
> > > > Thank you for updating your ratings.

---

### Official Review · Reviewer_1SA8 · 2025-07-05

**Clarity:** 3
**Significance:** 3
**Originality:** 2
**Rating:** 4
**Confidence:** 2

**Summary:**

This paper presents NeuroMP, a brain-inspired motion planning framework that models human planning processes through a two-stage structure: visuospatial perception and semantic-episodic synergistic decision-making. The proposed method is validated on multiple high-dimensional robotic arm tasks, showing superiority over existing classical and learning-based baselines.

**Questions:**

1.	Please clarify whether all baseline methods use the same selective sampling strategy. If not, please provide additional comparisons.
2.	Can you explain the difference between NeuroMP and BrainyMP in more detail?

**Ethical Concerns:**

["NO or VERY MINOR ethics concerns only"]

**Limitations:**

Yes, the authors have adequately addressed limitations in Appendix G.

**Paper Formatting Concerns:**

not any formatting issues.

**Quality:**

3

**Strengths And Weaknesses:**

Strengths:
1.	This paper is well-motivated, the brain-inspired method is very interesting; Each module has a clear purpose and is validated via ablation studies.
2.	The evaluation is comprehensive, covering multiple environments and comparing against a wide range of baselines.
3.	The paper is well-organized with clear figures and comprehensive methodology descriptions.
Weaknesses:
1.	The selective sampling strategy proposed in the paper wasn’t used for the baseline methods, which might make NeuroMP look better than it actually is. A fair comparison would make the results more convincing.
2.	Although NeuroMP introduces new components, its overall architecture and motivation share notable overlap with BrainyMP, particularly in brain-inspired design and selective sampling strategy. The distinction in contributions could be more clearly articulated.
3.	The main idea is built on existing GNN-based planning methods, and most of the improvements seem like incremental design. But the experimental results are solid and promising, so I recommend a borderline acceptance.

---

> ### Author Rebuttal · Authors · 2025-07-30
>
> We sincerely thank you for your positive assessment of our work, describing it as "well-motivated", "comprehensive", and "well-organized". We also appreciate your valuable comments and for pointing out the limitations in our work. Following your suggestions, we have provided detailed point-by-point replies below. We hope our revisions and clarifications adequately address your concerns.
>
> **Q1: Selective sampling**
>
> We sincerely appreciate your valuable comments. We agree that a fair comparison is crucial and thank you for raising this point regarding the selective sampling. The selective sampling strategy enables the robot to sample nodes from a guiding region with probability $\beta$, and uniformly from the entire configuration space with probability $1 - \beta$. The guiding region is defined as the circular area whose diameter is the line segment between the start and goal. This strategy helps optimize the solution space to some extent, thereby improving planning efficiency and path quality. Among the baselines, BrainyMP inherently employs a similar selective sampling strategy. Additionally, BIT* and LazySP utilize informed sampling strategies that also concentrate sampling within heuristic regions connecting the start and goal. To ensure a comprehensive and fair comparison specifically for the selective sampling component you highlighted, we have now integrated our selective sampling strategy into the core sampling procedures of RRT*, NEXT, GNN-Explorer, and GraphMP. The augmented results across six environments are summarized below :
> #### Table 1: Performance comparison on Stick3 on Link8
> |Method|SR|CK|PC|PCP|PT|SR|CK|PC|PCP|PT|
> |--|--|--|--|--|--|--|--|--|--|--|
> |GraphMP|0.97|7485.33|1.35|1.76|491.02|0.89|13222.98|10.59|13.80|1569.65|
> |**GraphMP w/**|0.97|7677.42|**1.33**|**1.67**|502.30|0.89|**12590.42**|**9.24**|**11.97**|**1520.51**|
> |GNN-Explorer|0.98|8805.60|2.02|2.23|687.95|0.91|10824.59|19.53|21.29|1872.64|
> |**GNN-Explorer w/**|0.98|9479.68|**1.93**|**2.14**|816.45|**0.93**|10152.95|**17.10**|**18.73**|**1767.66**|
> |NEXT|0.96|6380.69|1.22|1.77|586.63|0.41|13022.17|7.92|26.95|19494.83|
> |**NEXT w/**|0.96|6426.98|1.22|1.77|590.12|**0.42**|**12795.21**|7.97|**26.49**|**19146.13**|
> |RRT*|0.62|10106.18|1.36|6.38|235.55|0.41|9332.73|7.95|26.86|1254.93|
> |**RRT* w/**|**0.63**|10150.64|**1.33**|**6.22**|**224.17**|**0.47**|**8630.95**|**7.32**|**24.77**|**1167.35**|
>
> #### Table 2: Performance comparison on Ur5 and Kuka13
> |Method|SR|CK|PC|PCP|PT|SR|CK|PC|PCP|PT|
> |-----------|------|------|------|------|------|-----|------|------|------|------|
> |GraphMP|0.96|2706.42|8.78|10.05|266.55|0.99|645.51|11.42|12.68|124.71|
> |**GraphMP w/**|**0.98**|**2529.90**|**8.01**|**8.58**|**240.90**|0.99|**627.85**|**11.02**|**12.21**|**119.59**|
> |GNN-Explorer|0.98|3184.29|12.51|12.86|585.67|0.99|741.34|15.75|15.89|104.35|
> |**GNN-Explorer w/**|**0.99**|2844.05|**11.38**|**11.57**|478.06|**1.00**|681.97|**15.52**|**15.55**|108.35|
> |NEXT|0.37|6241.26|4.67|24.04|5014.84|0.61|4868.52|10.35|48.58|3830.19|
> |**NEXT w/**|0.37|**6172.81**|**4.63**|**23.83**|**4961.66**|**0.62**|4811.00|**10.24**|**45.89**|3956.78|
> |RRT*|0.39|3141.19|4.06|29.39|291.74|0.67|2981.80|9.15|30.47|269.91|
> |**RRT* w/**|**0.41**|3121.14|**4.03**|**28.86**|207.81|**0.69**|**2916.34**|**9.12**|**29.49**|**258.25**|
>
> #### Table 3: Performance comparison on Kuka2Arms and Kuka3Arms
> |Method|SR|CK|PC|PCP|PT|SR|CK|PC|PCP|PT|
> |-----------|-----|-----|----|----|------|---|-----|----|----|------|
> |GraphMP|0.98|581.96|11.19|11.77|88.94|0.61|1875.29|16.37|25.68|412.06|
> |**GraphMP w/**|0.98|**572.89**|**10.97**|**11.61**|**87.27**|**0.62**|**1855.84**|16.48|**25.44**|**406.63**|
> |GNN-Explorer|1.00|574.80|16.60|16.94|110.91|0.57|2880.16|20.62|33.40|764.84|
> |**GNN-Explorer w/**|1.00|597.39|**15.00**|**15.98**|161.26|0.57|**2865.41**|**20.21**|**33.01**|**745.60**|
> |NEXT|0.66|4637.57|10.26|48.74|3719.14|0.38|2141.68|15.39|37.25|4383.60|
> |**NEXT w/**|0.66|**4581.39**|**10.07**|**47.34**|3778.16|0.37|2188.96|**15.02**|37.45|**4316.56**|
> |RRT*|0.70|2810.07|9.69|29.87|203.75|0.25|2986.57|11.38|40.23|666.92|
> |**RRT* w/**|**0.72**|**2690.14**|**9.56**|**28.25**|**186.84**|0.25|3228.56|**10.76**|**39.94**|737.03|
>
> These results demonstrate that while selective sampling generally improves path quality and often efficiency, its impact varies with environmental complexity. In the Link8 to Kuka2Arms environments, collision checks and planning time are reduced. However, in the Stick3 environment, both increase slightly due to the high obstacle density, which introduces additional computational overhead in the heuristic region. Similarly, in the Kuka3Arms environment, planning efficiency decreases with the selective sampling strategy. This is likely due to the dense configuration of the three fixed-base manipulators, where sampling within the heuristic region often causes inter-arm collisions, increasing planning difficulty. Importantly, NeuroMP's superior performance reported in the original submission holds even when competing methods are augmented with this strategy, as evidenced by the comparative metrics.
>
> **Q2. The difference between NeuroMP and BrainyMP**
>
> We sincerely thank you for your valuable suggestions, which helped us improve the quality of our manuscript. Following your comments, we have clarified the differences between NeuroMP and BrainyMP in terms of motivation, technical design, and brain-inspired mechanisms. In addition, we have revised the introduction and related work sections to include a more detailed review of BrainyMP's core ideas and technical contributions.
>
> 1. Addressing BrainyMP's limitations by technical design
>
> BrainyMP addresses this limitation by integrating subgraph structures with node positional relationships to enhance the model’s ability to capture local graph representation patterns, explicitly encoding local structural information. However, BrainyMP still faces several limitations:
>
> (i) Inaccurate graph construction. The selector may mistakenly retain collision-prone edges or eliminate valid ones, which adversely affects heuristic estimation.
>
> (ii) Limited structural reasoning. Under weakly connected graph structures, although subgraphs can partially enhance local–global relationships, their ability to capture global information is constrained by the number and coverage of subgraphs, limiting the overall structural reasoning capability.
>
> (iii) Computational complexity and redundancy. The process of extracting subgraphs incurs a high computational cost, and different subgraphs may contain redundant information, reducing efficiency.
>
> To address these limitations, we propose NeuroMP with the following key contributions:
>
> (i) To improve collision probability prediction, the Perceptive Segment Selector adopts a gated dual-branch transformer architecture that efficiently identifies and reasons about environmental patterns and their interrelations.
>
> (ii) To enhance heuristic estimation in weakly connected graphs, the Global Alignment Heuristic incorporates spectral features to impose global topological constraints. Additionally, a dual-channel collaborative learning strategy (A2FL) is introduced to leverage the complete graph to assist in training on weakly connected graphs.
>
> (iii) Spectral features help avoid incomplete coverage, and compared to the introduction of subgraphs, they reduce computational complexity.
>
> 2. Distinct brain-inspired principles
>
> While both works draw inspiration from neuroscience, the level of abstraction and specific cognitive processes modeled differ significantly.
>
> BrainyMP draws inspiration from cognitive neuroscience, specifically the Tolman–Eichenbaum Model (TEM) of spatial relational memory, which emphasizes how the brain integrates sensory observations with relational structures to perform sensory inference. Based on this concept, BrainyMP enhances its model structure by incorporating subgraph-based representations to encode local relational patterns.
>
> In contrast, NeuroMP is grounded in a higher-level cognitive framework, namely the two-stage Perception–Decision model observed in human planning and reasoning. The first stage, mimicking the visuospatial perception process, informs the design of the Perceptive Segment Selector, which captures environmental features and spatial relations to guide graph construction. The second stage, motivated by semantic–episodic synergistic decision-making, guides the design of both the heuristic model architecture and the learning strategy, enabling more effective structural reasoning and improved generalization in weakly connected graphs. Overall, compared to BrainyMP, NeuroMP leverages a higher-level cognitive framework to address the challenges of inaccurate graph construction and weak graph reasoning.

---

### Note · Authors · 2025-08-14

We sincerely thank all reviewers and chairs for their time and effort. We greatly appreciate that the reviewers have acknowledged our novelty, motivation, clear presentation, and comprehensive experiments.

We address the two major concerns raised by multiple reviewers: generalization of real-world environments and contribution of brain-inspired design. For all other issues, detailed responses are provided under each reviewer’s comments.

**1.Generalization of real-world environments**

We present new results on the real-world CSM dataset. The dataset was collected by Sturtevant et al. using drones from 30 real city maps with marked obstacles represented as binary images. These maps present significantly higher complexity than synthetic environments: irregular obstacle shapes, narrow alleyways, and dense urban structures that challenge traditional planners. These results demonstrate that NeuroMP not only achieves competitive success rates but also requires the fewest collision checks and the shortest planning time among all baselines.

Our method requires further validation in real-world deployment scenarios. For future work, we plan to evaluate our approach under more realistic settings, including robotic arm manipulation and mobile robot navigation.

**2.Contribution of brain-inspired design**

Compared with the SOTA method BrainyMP, inspired by the human perception–decision model, NeuroMP enhances the perception of dangerous edges and the prediction of heuristics, addressing the challenges of inaccurate graph construction and weak graph reasoning.

Brain-inspired methods for motion planning remain in an exploratory stage, but these methods represent a novel and promising direction. Prior studies have already begun to validate the potential benefits of learning from the brain. We also believe that drawing inspiration from human brains holds considerable promise for enhancing robotic decision-making and planning.

---
Following the reviewers’ suggestions, we will expand the literature review, simplify the brain-inspired expressions to better highlight the methodological description, and incorporate additional experimental results in the revision. Furthermore, we will strengthen the analysis and discussion of performance improvements to underscore the contribution of NeuroMP. In addition, we will enrich the discussion of limitations and future work, such as static environment constraints and brain-inspired analogies.

---

### Decision · Program_Chairs · 2025-09-17

**Decision:**

Accept (poster)

**Comment:**

The paper proposes a path-planning mechanism for robotics. The perception module builds a graph that approximates the possible safe paths and transitions.The decision module assigns heuristic values to the graph, allowing an efficient A*-style search. The method does not depend on specific features in the perception stage for defining the search space. Instead, the authors employ a differentiable variant of A*, which assumes, among other things, that the nodes are vectors. The submission claims that the heuristic helps compensate for errors in the graph prediction, avoiding dangerous edges.

The reviewers find the work technically solid and the manuscript well written. There was some discussion of novelty. The consensus is that the work is novel and interesting, though gaps in the scholarship caused some hesitation. For one reviewer, it was important to clarify the differences with respect to BrainyMP.

The sampling mechanism was also discussed at length. Since the representation depends on the samples, questions arose regarding the quality of the resulting graph. The authors addressed the general issue of sampling for graph construction. The experiments in the submission, together with additional results posted during the discussion, persuaded the reviewers of the soundness of this aspect. A more detailed discussion of the complexity of the benchmarks also helped to convince the reviewers that the paper deserves acceptance.

The authors must improve the scholarship of the paper. They should clarify the differences with prior work, update the experimental results, and discuss the significance of the new experiments.